# Semi-supervised Multi-label Learning with Balanced Binary Angular Margin Loss

**Ximing Li**[1,2]  **Silong Liang**[1,2]  **Changchun Li**[1,2,*]  **Pengfei Wang**[3,4]  **Fangming Gu**[1,2]

[1]College of Computer Science and Technology, Jilin University, China
[2]Key Laboratory of Symbolic Computation and Knowledge Engineering of Ministry of Education, Jilin University, China
[3]Computer Network Information Center, Chinese Academy of Sciences, China
[4]University of Chinese Academy of Sciences, Chinese Academy of Sciences, China
`liximing86@gmail.com, changchunli93@gmail.com, liangsl23@mails.jlu.edu.cn,`
`pfwang@cnic.cn, gufm@jlu.edu.cn`

## Abstract

Semi-supervised multi-label learning (SSMLL) refers to inducing classifiers using a small number of samples with multiple labels and many unlabeled samples. The prevalent solution of SSMLL involves forming pseudo-labels for unlabeled samples and inducing classifiers using both labeled and pseudo-labeled samples in a self-training manner. Unfortunately, with the commonly used binary type of loss and negative sampling, we have empirically found that learning with labeled and pseudo-labeled samples can result in the variance bias problem between the feature distributions of positive and negative samples for each label. To alleviate this problem, we aim to balance the variance bias between positive and negative samples from the perspective of the feature angle distribution for each label. Specifically, we extend the traditional binary angular margin loss to a balanced extension with feature angle distribution transformations under the Gaussian assumption, where the distributions are iteratively updated during classifier training. We also suggest an efficient prototype-based negative sampling method to maintain high-quality negative samples for each label. With this insight, we propose a novel SSMLL method, namely **S**emi-**S**upervised **M**ulti-**L**abel **L**earning with **B**alanced **B**inary **A**ngular **M**argin loss ($\mathbf{S}^2\mathbf{ML}^2$**-BBAM**). To evaluate the effectiveness of $\mathrm{S}^2\mathrm{ML}^2$-BBAM, we compare it with existing competitors on benchmark datasets. The experimental results validate that $\mathrm{S}^2\mathrm{ML}^2$-BBAM can achieve very competitive performance.

## 1 Introduction

Multi-label learning (MLL) refers to the classification problem where each training sample can be associated with multiple labels [1]. For example, in text categorization, a text can involve a certain number of topics simultaneously [2, 3]; and in image annotation, an image can contain multiple objects of interest in one scene [4, 5]. Compared with single-label learning, MLL is a more prevalent paradigm in real-world scenarios, and it has been widely used in many applications such as information retrieval [6, 7] and recommendation systems [8, 9].

---

*Corresponding author.

38th Conference on Neural Information Processing Systems (NeurIPS 2024).

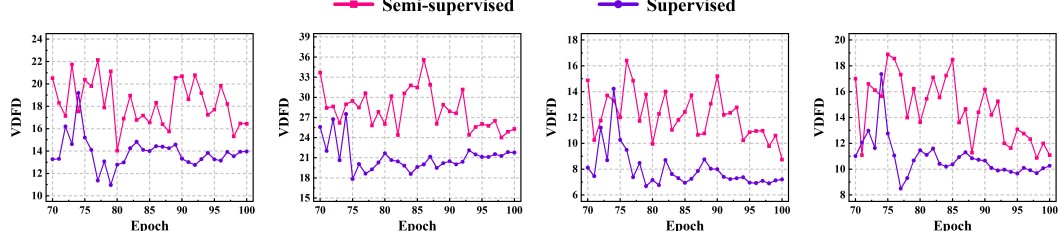

Figure 1: The variance difference between feature distributions (VDFD) of positive and negative samples computed in semi-supervised and supervised manners across labels $\{6, 7, 14, 17\}$ of *VOC2012*.

Despite the successful application of MLL, the competitive performance of most MLL methods heavily depends on the large volume of training samples with precise supervision [4, 10, 11]. Unfortunately, it is expensive to manually annotate each sample, so it is naturally time-consuming to collect loads of labeled training samples. Accordingly, the community has turned to alternative candidates to MLL, and raised the question of whether one can induce robust MLL classifiers with a small number of labeled samples and a large number of unlabeled samples, which are cheaper to collect. This concept gives birth to the emerging research topic of **s**emi-**s**upervised **m**ulti-**l**abel **l**earning (**SSMLL**), and many attempts have been recently proposed [12, 13, 14, 15, 16, 17].

Generally, the topic of SSMLL, as its name suggests, is in parallel inherited from semi-supervised learning (SSL) and MLL. The current prevalent ideas are estimating pseudo-labels of unlabeled samples with SSL techniques and inducing MLL classifiers with both labeled and pseudo-labeled samples in a self-training manner. Following the prior arts [18, 19], the binary kind of losses, *e.g.* binary cross-entropy loss and asymmetric loss [20], are commonly used to optimize MLL classifiers, where those are equivalent to optimizing the binary loss between the positive and negative samples for each label. To alleviate the imbalanced issue between positive and negative samples, especially for the scenarios with massive labels, the negative sampling tricks are often employed [21, 22, 23]. Unfortunately, in our preliminary experiments, we found such training paradigms suffer from the **variance bias** problem by using the labeled and pseudo-labeled samples in the context of SSMLL, since it is difficult to guarantee estimating accurate pseudo-labels. To be specific, the problem implies that for each label, in SSMLL the variance difference between feature distributions of positive and negative samples is often larger than the ones in fully supervised learning, as illustrated in Fig.1. In this situation, each trained binary boundary tends to keep away from the Bayesian optimal one, resulting in performance degradation.

To tackle this problem, we propose a novel SSMLL method, namely **S**emi-**S**upervised **M**ulti-**L**abel **L**earning with **B**alanced **B**inary **A**ngular **M**argin loss ($\text{S}^2\text{ML}^2$-BBAM). The basic insight of $\text{S}^2\text{ML}^2$-BBAM is to balance the variance bias between positive and negative samples from the perspective of the feature angle distribution for each label. To be specific, we extend the binary angular margin (BAM) loss, which measures the prediction loss by using the angle between the feature and binary boundary for each label. We suppose that for each label these feature angles of positive and negative samples are drawn from label-specific "positive" and "negative" Gaussian distributions, which are estimated by employing both labeled and pseudo-labeled samples during classifier training. Therefore, we can apply some linear Gaussian transformations over these feature angle distributions, so as to balance the variance bias between positive and negative samples for each label. Upon this idea, we design a new balanced binary angular margin (BBAM) loss and construct a novel $\text{S}^2\text{ML}^2$-BBAM method based on the designed BBAM loss and self-training manner. We also suggest an efficient prototype-based negative sampling method to maintain high-quality negative samples for each label. We evaluate the proposed $\text{S}^2\text{ML}^2$-BBAM by comparing the most recent competitors on benchmark datasets. Experimental results indicate the superior performance of $\text{S}^2\text{ML}^2$-BBAM.

In summary, the main contributions of this paper are listed as follows:

- We develop a novel SSMLL method, namely $\text{S}^2\text{ML}^2$-BBAM, by balancing the variance bias between positive and negative samples from the perspective of the feature angle distribution for each label.

- We design a new BBAM loss by extending the traditional binary angular margin loss with feature angle distribution transformations under the Gaussian assumption, and suggest an efficient prototype-based negative sampling method to maintain high-quality negative samples for each label.

- We construct extensive experiments to evaluate $S^2ML^2$-BBAM, and experimental results demonstrate the effectiveness of $S^2ML^2$-BBAM.

## 2 Formulation and Analysis

### 2.1 Problem Formulation

By convention, we use $\mathbf{x}$ to denote the sample feature vector and $\mathbf{y} \in \{0,1\}^K$ the label indicator vector of $K$ pre-defined classes, where $0/1$ implies a sample is irrelevant/relevant to the category. In the task of SSMLL, we are formally given a collection of training samples $\mathcal{D} = \{\mathcal{D}_l, \mathcal{D}_u\}$, where $\mathcal{D}_l = \{(\mathbf{x}_i^l, \mathbf{y}_i^l)\}_{i=1}^{i=N_l}$ and $\mathcal{D}_u = \{\mathbf{x}_j^u\}_{j=1}^{j=N_u}$ are the collections of $N_l$ labeled and $N_u$ unlabeled samples, respectively. The goal of SSMLL is to induce a classifier $f_{\mathbf{W}}(\mathbf{x})$, parameterized by $\mathbf{W}$, from $\mathcal{D}$ and use the classifier $f_{\mathbf{W}}(\mathbf{x})$ to predict the label indicator vectors for future samples.

Broadly speaking, the classifier $f_{\mathbf{W}}(\mathbf{x})$ typically consists of a backbone encoder and a classification layer, parameterized by $\mathbf{W}^e$ and $\mathbf{W}^c$, respectively (*i.e.* $\mathbf{W} = \{\mathbf{W}^e, \mathbf{W}^c\}$). Specifically, the backbone encoder transforms any original feature vector $\mathbf{x}$ into a more discriminative latent feature $\mathbf{z} = f_{\mathbf{W}^e}(\mathbf{x})$; the classification layer applies $\mathbf{z}$ to generate its corrsponding predictive logits $\mathbf{p} = f_{\mathbf{W}^c}(\mathbf{z})$. Given an SSMLL training dataset $\mathcal{D}$, the classifier $f_{\mathbf{W}}(\mathbf{x})$ is commonly optimized by minimizing the following generic self-training objective concerning $\mathbf{W}$ on $B_l$-sized labeled and $B_u$-sized unlabeled batches:

$$\mathcal{L}(\mathbf{W}) = \frac{1}{B_l K} \sum_{i=1}^{B_l} \sum_{k=1}^{K} \ell(p_{ik}^l, y_{ik}^l) + \frac{\lambda}{B_u K} \sum_{i=1}^{B_u} \sum_{k=1}^{K} \ell(p_{ik}^u, y_{ik}^u), \tag{1}$$

where $\ell(\cdot, \cdot)$ is a binary loss function; $\lambda$ is the coefficient parameter; $\mathbf{p}_i^l = f_{\mathbf{W}}(\mathbf{x}_i^l)$ and $\mathbf{p}_i^u = f_{\mathbf{W}}(\mathbf{x}_i^u)$ are the predictive logits of labeled and unlabeled samples, respectively; $\mathbf{y}_i^u$ is the pseudo-label of unlabeled samples induced from its current classifier prediction $\mathbf{p}_i^u$.

### 2.2 How Variance Bias Affects the Performance

As shown in Fig.1, we have observed that the generic self-training objective of SSMLL may suffer from the variance bias problem. Here, we discuss how it will affect the classification performance. We treat SSMLL as $K$ independent semi-supervised binary classification (SSBC) tasks. For each SSBC task, let $\{(\mathbf{x}_i, y_i^*)\} \cup \{\mathbf{x}_i\}$ be the training data, where $\mathbf{x} \in \mathbb{R}^d$ and $y^* \in \{-1, +1\}$ is the ground-truth label. Besides, let $\widehat{y} \in \{-1, +1\}$ be the pseudo-label. For clarity and conciseness, we study the SSBC training data drawn from a mixture Gaussian distribution $\mathcal{P}^*$, which can be defined by the following distribution over $(\mathbf{x}, y) \in \mathbb{R}^d \times \{\pm 1\}$:

$$y = \begin{cases} +1, & p = \alpha, \\ -1, & p = 1 - \alpha, \end{cases} \qquad \mathbf{x} \sim \begin{cases} \mathcal{N}(\boldsymbol{\mu}, \boldsymbol{\Sigma}_+^2) & \text{if } y = +1; \\ \mathcal{N}(-\boldsymbol{\mu}, \boldsymbol{\Sigma}_-^2) & \text{if } y = -1, \end{cases} \tag{2}$$

where $\alpha$ is the prior probability of class "$+1$", $\boldsymbol{\mu} = \{\mu_1, \ldots, \mu_d\}^\top$, $\boldsymbol{\Sigma}_+ = \text{diag}(\{\sigma_+^{(1)}, \ldots, \sigma_+^{(d)}\})$, $\boldsymbol{\Sigma}_- = \text{diag}(\{\sigma_-^{(1)}, \ldots, \sigma_-^{(d)}\})$, $\mu_i, \sigma_-^{(i)}, \sigma_+^{(i)} > 0 \ \forall i \in [d]$, and $\sum_{i=1}^d (\sigma_+^{(i)})^2 : \sum_{i=1}^d (\sigma_-^{(i)})^2 = 1 : M^2$ with $M > 0, M \neq 1$. We concentrate on analyzing the effect of the variance proportion $M$ of the distribution $\mathcal{D}^*$ on the performance of the linear model $f_{ssl}(\mathbf{x}) = \text{sign}(\langle \mathbf{w}, \mathbf{x} \rangle + b)$, where the parameters $\mathbf{w} \in \mathbb{R}^d, b \in \mathbb{R}$, and $\text{sign}(t)$ evaluates to $+1$ if scalar $t \geq 0$ and to $-1$ otherwise. For simplicity, we denote

$$\mathcal{R}(f, +1) = \mathbb{E}_{(\mathbf{x},y) \sim \mathcal{P}^*}[\mathbb{1}(f(\mathbf{x}) = -1)|y = +1], \ \mathcal{R}(f, -1) = \mathbb{E}_{(\mathbf{x},y) \sim \mathcal{P}^*}[\mathbb{1}(f(\mathbf{x}) = +1)|y = -1],$$

where $\mathbb{1}(t)$ is the indicator function that takes 1 where t is true and 0 otherwise. We have the following theorems, whose proof can be found in the Appendix A.

**Theorem 2.1.** *Given an SSBC dataset with pseudo-labels $\mathcal{S} = \{(\mathbf{x}_i, y_i)\} = \{(\mathbf{x}_i, y_i^*)\} \cup \{(\mathbf{x}_i, \widehat{y}_i)\}$, the optimal linear classifier $f_{ssl}$ minimizing the average standard classification error is given by:*

$$f_{ssl} = \arg\min_{f} \mathbb{E}_{(\mathbf{x},y)\sim\mathcal{S}}[\mathbb{1}(f(\mathbf{x}) \neq y)]. \tag{3}$$

*When $M > 1$, it has the intra-class standard classification errors for the two classes :*

$$\mathcal{R}(f_{ssl}, +1) = \Phi\big(A - M\sqrt{A^2 + q(M, \alpha, \epsilon_-, \epsilon_+)}\big),$$
$$\mathcal{R}(f_{ssl}, -1) = \Phi\big(-M \cdot A + \sqrt{A^2 + q(M, \alpha, \epsilon_-, \epsilon_+)}\big),$$

*and when $M < 1$, they are given by:*

$$\mathcal{R}(f_{ssl}, +1) = \Phi\big(A + M\sqrt{A^2 + q(M, \alpha, \epsilon_-, \epsilon_+)}\big),$$
$$\mathcal{R}(f_{ssl}, -1) = \Phi\big(-M \cdot A - \sqrt{A^2 + q(M, \alpha, \epsilon_-, \epsilon_+)}\big),$$

*where $\Phi(\cdot)$ is the cumulative distribution function (c.d.f.) of standard Gaussian distribution $\mathcal{N}(0,1)$, $A = \frac{2\mu}{(M^2-1)\Sigma}$, $q(M, \alpha, \epsilon_-, \epsilon_+) = \frac{2\log M + 2C}{M^2 - 1}$, $C = \log\big(\frac{\alpha(2-\epsilon_- - 2\epsilon_+)}{(1-\alpha)(2-2\epsilon_- -\epsilon_+)}\big)$, $\mu = \sum_{i=1}^{i=d} \mu_i$, $\Sigma = \sqrt{\sum_{i=1}^{i=d}(\sigma_+^{(i)})^2}$, and $\{\epsilon_-, \epsilon_+\}$ are the proportions of negative instances being treated as positive ones and positive instances being treated as negative ones within pseudo-labels, respectively. If $\sum_{i=1}^{d}(\sigma_+^{(i)})^2 = \sum_{i=1}^{d}(\sigma_-^{(i)})^2$, i.e. $M = 1$, the intra-class standard classification errors for the two classes can be expressed as follows:*

$$\mathcal{R}(f_{ssl}, +1) = \Phi\big(\frac{-2\mu^2 - C\Sigma^2}{2\mu\Sigma}\big), \quad \mathcal{R}(f_{ssl}, -1) = \Phi\big(\frac{-2\mu^2 + C\Sigma^2}{2\mu\Sigma}\big).$$

Following [24, 25, 26], We employ *variance of class-wise accuracy* (VCA) to quantitatively measure the model fairness and present the definition of VCA below.

**Definition 2.2.** (VCA) Given a classifier $f : \mathcal{X} \to \mathcal{Y}$ where $\mathcal{Y} = \{1, 2, 3, \cdots, K\}$, the variance of class-wise accuracy of $f$ is defined as $VCA(f) = \frac{1}{K}\sum_{i=1}^{K}(p(i) - \bar{p})$, where $p(i) = \mathbb{P}[f(\mathbf{x}) = i | y = i] = 1 - \mathbb{P}[f(\mathbf{x}) \neq i | y = i]$ and $\bar{p} = \frac{1}{K}\sum_{i=1}^{K}p(i)$.

**Theorem 2.3.** *Given an trained linear SSBC model $f_{ssl}$ in Eq.(3), the variance of class-wise accuracy $VCA(f_{ssl})$ is increasing when $M \to \infty$ for $M > 1$ and $M \to 0$ for $M < 1$. Suppose $\log\big(\frac{\alpha(2-\epsilon_- - 2\epsilon_+)}{(1-\alpha)(2-2\epsilon_- -\epsilon_+)}\big) = 0$, then when $M = 1$, $\mathcal{R}(f_{ssl}, +1) = \mathcal{R}(f_{ssl}, -1)$ and $VCA(f_{ssl}) = 0$.*

*Remark* 2.4. According to Theorem 2.3, the bigger or smaller value of $M$ will result in the increase of the variance of class-wise accuracy $VCA(f_{ssl})$, which implies that the SSBC classifier $f_{ssl}$ induced by Eq.(3) is unfair. Note that $M$ is the variance proportion of feature distributions of positive and negative samples as defined in (2). Therefore, to improve the fairness of the induced classifier, we propose to balance the variance bias of positive and negative samples for each label from the feature angle distribution perspective, leading to our $S^2ML^2$-BBAM.

# 3 Proposed $S^2ML^2$-BBAM Method

In this section, we introduce the proposed SSMLL method named $\mathbf{S^2ML^2}$-**BBAM**.

## 3.1 Overview

Generally, our $S^2ML^2$-BBAM is built on the generic self-training objective of SSMLL formulated by Eq.1. Specifically, we propose a novel **B**alanced **B**inary **A**ngular **M**argin (**BBAM**) loss $\ell_{\text{BBAM}}(\cdot, \cdot)$, aiming to balance the variance bias of positive and negative samples for each label from the feature angle distribution perspective with the Gaussian assumption. By applying our proposed BBAM loss to the generic SSMLL self-training objective in Eq.1, the objective of $S^2ML^2$-BBAM can be formulated as follows:

$$\mathcal{L}(\mathbf{W}) = \frac{1}{B_l K}\sum_{i=1}^{B_l}\sum_{k=1}^{K}\beta_{ik}\ell_{\text{BBAM}}(p_{ik}^l, y_{ik}^l) + \frac{\lambda}{B_u K}\sum_{i=1}^{B_u}\sum_{k=1}^{K}\beta_{ik}\ell_{\text{BBAM}}(p_{ik}^u, y_{ik}^u), \tag{4}$$

where

$$\beta_{ik} = \begin{cases} 1 & \text{if } (\mathbf{x}_i, \mathbf{y}_i) \in \Omega_k; \\ 1 & \text{if } y_{ik} = 1; \\ 0 & \text{otherwise}, \end{cases} \quad \forall k \in [K], \ \forall i \in [N_l] \text{ or } [N_u],$$

and $\{\Omega_k\}_{k=1}^{k=K}$ denotes high-quality negative sample sets constructed by negative sampling.

Here, pseudo-labels of unlabeled data $\{\mathbf{y}_i^u\}_{i=1}^{i=N_u}$ are produced by employing the Class-Aware Pseudo-labeling (CAP) trick [16], which drives their label distribution towards the prior one that is estimated with the labeled samples. Specifically, given the current classifier predictions $\{\mathbf{p}_i^u\}_{i=1}^{i=N_u}$ of unlabeled samples, $\{\mathbf{y}_i^u\}_{i=1}^{i=N_u}$ are given by:

$$
y_{ik}^u = \begin{cases} 1 & \text{if } p_{ik}^u >= \delta_k; \\ 0 & \text{if } p_{ik}^u <= \gamma_k; \qquad \forall k \in [K], \ \forall i \in [N_u], \\ -1 & \text{otherwise,} \end{cases} \tag{5}
$$

where the class-aware thresholds $\{\delta_k\}_{k=1}^{k=K}$ and $\{\gamma_k\}_{k=1}^{k=K}$ are calculated by solving the equations:

$$
\begin{cases} \frac{\sum_{i=1}^{N_u} \mathbb{1}(p_{ik}^u >= \delta_k)}{N_u} = \frac{\sum_{i=1}^{N_l} \mathbb{1}(y_{ik}^l = 1)}{N_l}, \\ \\ \frac{\sum_{i=1}^{N_u} \mathbb{1}(p_{ik}^u <= \gamma_k)}{N_u} = \frac{\sum_{i=1}^{N_l} \mathbb{1}(y_{ik}^l = 0)}{N_l}, \end{cases} \qquad \forall k \in [K], \ \forall i \in [N_u],
$$

and $y_{ik}^u = -1$ means that it will not be used for the classifier training.

## 3.2 BBAM loss

In this section, we introduce the proposed BBAM loss. As its name suggests, our BBAM loss is extended from the **B**inary **A**ngular **M**argin (BAM) loss, which measures the label-specific prediction risk by using the angle between the latent feature and boundary. Formally, for a training sample $(\mathbf{x}_i, \mathbf{y}_i)$, the BAM loss can be formulated as:

$$
\ell_{\text{BAM}}(p_{ik}, y_{ik}) = \begin{cases} -\log\left(\frac{1}{1+e^{-s*(p_{ik}-m)}}\right) & \text{if } y_{ik} = 1; \\ \\ -\log\left(1 - \frac{1}{1+e^{-s*(p_{ik}-m)}}\right) & \text{if } y_{ik} = 0, \end{cases} \tag{6}
$$

where $p_{ik} = \cos(\theta_{ik}) = \frac{\mathbf{z}_i^\top \mathbf{W}_k^c}{\|\mathbf{z}_i\|_2 \|\mathbf{W}_k^c\|_2}$, $\|\cdot\|_2$ is the $\ell_2$-norm of vectors; $\mathbf{z}_i$ and $\mathbf{W}_k^c$ denote the latent feature of sample $i$ and the weight vector of the classification layer for category $k$, respectively; $\theta_{ik}$ is the angle between $\mathbf{z}_i$ and $\mathbf{W}_k^c$; $s$ and $m$ are the parameters used to control the rescaled norm and magnitude of cosine margin, respectively.

Reviewing the BAM loss in Eq.6, one can observe that it calculates the loss by employing the label angles of samples for each category. We consider that its trained binary boundary tends to deviate from the Bayesian optimal one for each category in SSMLL, where for most categories, the differences between feature distribution variances of corresponding positive and negative samples are much larger than ones in fully supervised learning. To address this issue, for each category $k$, we suppose that label angles of its positive samples and ones of its negative samples are drawn from a label-specific "positive" Gaussian distribution $\mathcal{N}(\mu_k^{(p)}, (\sigma_k^2)^{(p)})$ and a label-specific "negative" one $\mathcal{N}(\mu_k^{(n)}, (\sigma_k^2)^{(n)})$, respectively. According to the properties of Gaussian distribution, we can easily transfer them into ones $\mathcal{N}(\mu_k^{(p)}, \hat{\sigma}_k^2)$ and $\mathcal{N}(\mu_k^{(n)}, \hat{\sigma}_k^2)$ with balanced variance $\hat{\sigma}_k^2 = \frac{(\sigma_k^2)^{(p)}+(\sigma_k^2)^{(n)}}{2}$, by performing the following Gaussian linear transformations on those label angles:

$$
\psi_k^{(p)}(\theta_{ik}) = a_k^{(p)}\theta_{ik} + b_k^{(p)}, \quad \psi_k^{(n)}(\theta_{ik}) = a_k^{(n)}\theta_{ik} + b_k^{(n)},
$$

$$
a_k^{(p)} = \frac{\hat{\sigma}_k}{\sigma_k^{(p)}}, \quad b_k^{(p)} = (1 - a_k^{(p)})\mu_k^{(p)}, \quad a_k^{(n)} = \frac{\hat{\sigma}_k}{\sigma_k^{(n)}}, \quad b_k^{(n)} = (1 - a_k^{(n)})\mu_k^{(n)}, \quad \forall k \in [K]. \tag{7}
$$

With these linear transformation pairs $\{(\psi_k^{(p)}(\cdot), \psi_k^{(n)}(\cdot))\}$, for each category, label angles of both positive and negative samples can be refined into ones drawn from balanced angular distributions with one same variance, *e.g.*

$$
\psi_k^{(p)}(\theta_{ik}) \sim \mathcal{N}(\mu_k^{(p)}, \hat{\sigma}_k^2) \quad \text{if } y_{ik} = 1; \quad \psi_k^{(n)}(\theta_{ik}) \sim \mathcal{N}(\mu_k^{(n)}, \hat{\sigma}_k^2) \quad \text{if } y_{ik} = 0.
$$

Accordingly, the BAM loss in Eq.6 can be rewritten as the following BBAM loss:

$$\ell_{\text{BBAM}}(p_{ik}, y_{ik}) = \begin{cases} -\log\big(\frac{1}{1+e^{-s*(\cos(\psi_k^{(p)}(\theta_{ik}))-m)}}\big) & \text{if } y_{ik} = 1; \\ \\ -\log\big(1 - \frac{1}{1+e^{-s*(\cos(\psi_k^{(n)}(\theta_{ik}))-m)}}\big) & \text{if } y_{ik} = 0. \end{cases} \tag{8}$$

**Estimating label angle variances.** As mentioned above, we concentrate on estimating label-specific "positive" and "negative" angular distributions, *i.e.* $\{\mathcal{N}(\mu_k^{(p)}, (\sigma_k^2)^{(p)})\}_{k=1}^{k=K}$ and $\{\mathcal{N}(\mu_k^{(n)}, (\sigma_k^2)^{(n)})\}_{k=1}^{k=K}$, for each category whose draws are the angles between its label prototype $\mathbf{c}_k$ and latent features of its corresponding positive and negative samples, respectively. Here, we approximate $\{(\mu_k^{(p)}, (\sigma_k^2)^{(p)})\}_{k=1}^{k=K}$, $\{(\mu_k^{(n)}, (\sigma_k^2)^{(n)})\}_{k=1}^{k=K}$, and $\{\mathbf{c}_k\}_{k=1}^{k=K}$ with labeled and pseudo-labeled samples per-epoch.

For convenience, we denote $\mathfrak{D} = \{(\mathbf{z}_i, \mathbf{y}_i)\}_{i=1}^{i=N_l+N_u} = \{(\mathbf{z}_i^l, \mathbf{y}_i^l)\}_{i=1}^{i=N_l} \cup \{(\mathbf{z}_i^u, \mathbf{y}_i^u)\}_{i=1}^{i=N_u}$ as the couple set of latent features and labels or pseudo-labels of training samples $\mathcal{D}$ in the current epoch. We calculate label prototypes $\{\mathbf{c}_k\}_{k=1}^{k=K}$ by averaging latent features of positive samples in $\mathfrak{D}$ as:

$$\mathbf{c}_k = \frac{\sum_{i=1}^{N_l+N_u} \mathbb{1}(y_{ik} = 1)\mathbf{z}_i}{\sum_{i=1}^{N_l+N_u} \mathbb{1}(y_{ik} = 1)}, \ \forall k \in [K]. \tag{9}$$

Consequently, the label angles between label prototypes and latent features of samples are given by:

$$\phi_{ik} = \arccos\big(\frac{\mathbf{z}_i^\top \mathbf{c}_k}{\|\mathbf{z}_i\|_2 \|\mathbf{c}_k\|_2}\big), \ \forall k \in [K], \ \forall i \in [N_l + N_u],$$

Accordingly, the estimations of $\{(\mu_k^{(p)}, (\sigma_k^2)^{(p)})\}_{k=1}^{k=K}$ and $\{(\mu_k^{(n)}, (\sigma_k^2)^{(n)})\}_{k=1}^{k=K}$ based on the current negative sample sets $\{\Omega_k\}_{k=1}^{k=K}$ can be formulated as:

$$\mu_k^{(p)} = \frac{\sum_{i=1}^{N_l+N_u} \mathbb{1}(y_{ik} = 1)\phi_{ik}}{\sum_{i=1}^{N_l+N_u} \mathbb{1}(y_{ik} = 1)}, \qquad (\sigma_k^2)^{(p)} = \frac{\sum_{i=1}^{N_l+N_u} \mathbb{1}(y_{ik} = 1)(\phi_{ik} - \mu_k^{(p)})^2}{\sum_{i=1}^{N_l+N_u} \mathbb{1}(y_{ik} = 1) - 1},$$

$$\mu_k^{(n)} = \frac{\sum_{i=1}^{N_l+N_u} \beta_{ik}\mathbb{1}(y_{ik} = 0)\phi_{ik}}{\sum_{i=1}^{N_l+N_u} \beta_{ik}\mathbb{1}(y_{ik} = 0)}, \qquad (\sigma_k^2)^{(n)} = \frac{\sum_{i=1}^{N_l+N_u} \beta_{ik}\mathbb{1}(y_{ik} = 0)(\phi_{ik} - \mu_k^{(n)})^2}{\sum_{i=1}^{N_l+N_u} \beta_{ik}\mathbb{1}(y_{ik} = 0) - 1}. \tag{10}$$

Besides, to avoid the misleading effect of false positive or negative samples, we also employ moving average with a learning rate $\rho$ over $\{(\mu_k^{(p)}, (\sigma_k^2)^{(p)})\}_{k=1}^{k=K}$, $\{(\mu_k^{(n)}, (\sigma_k^2)^{(n)})\}_{k=1}^{k=K}$, and $\{\mathbf{c}_k\}_{k=1}^{k=K}$.

### 3.3 Negative Sampling

For efficiency, we suggest a prototype-based negative sampling method. Specifically, for each label, we tend to select those negative samples that are more similar to its positive samples, because they are more difficult to discriminate and would be more informative for the classifier training [21, 22]. To achieve this, for each category, we measure similarity scores of negative samples based on label prototypes $\{\mathbf{c}_k\}_{k=1}^{k=K}$, and construct the nearest neighbor negative sample sets $\{\widetilde{\Omega}_k\}_{k=1}^{k=K}$ as:

$$\widetilde{\Omega}_k = \{(\mathbf{x}_i, \mathbf{y}_i)|d(\mathbf{z}_i, \mathbf{c}_k) \in \text{Rank}(\{d(\mathbf{z}_i, \mathbf{c}_k)\}_{(\mathbf{x}_i, \mathbf{y}_i) \in \widehat{\Omega}_k}), (\mathbf{x}_i, \mathbf{y}_i) \in \widehat{\Omega}_k\} \quad \forall k \in [K],$$

where $d(\cdot)$ is the vector distance (*e.g.* cosine distance), $\text{Rank}(\cdot)$ outputs a set of samples with the top-$M$ minmum distance values; and $\{\widehat{\Omega}_k\}_{k=1}^{k=K}$ is the negative sample set of category $k$ defined as:

$$\widehat{\Omega}_k = \{(\mathbf{x}_i^l, \mathbf{y}_i^l)|(\mathbf{x}_i^l, \mathbf{y}_i^l) \in \mathcal{D}_l, y_{ik}^l = 0\} \cup \{(\mathbf{x}_i^u, \mathbf{y}_i^u)|\mathbf{x}_i^u \in \mathcal{D}_u, y_{ik}^u = 0\}.$$

Accordingly, the final negative sample sets $\{\Omega_k\}_{k=1}^{k=K}$ are generated by:

$$\Omega_k = \{(\mathbf{x}_i, \mathbf{y}_i)|(\mathbf{x}_i, \mathbf{y}_i) \sim \text{Uniform}(\widetilde{\Omega}_k)\} \quad \forall k \in [K], \tag{11}$$

with size $\{|\Omega_k| = \eta N_k\}_{k=1}^{k=K}$, where $N_k = \sum_{i=1}^{N_l} \mathbb{1}(y_{ik}^l = 1) + \sum_{i=1}^{N_u} \mathbb{1}(y_{ik}^u = 1)$, $\eta$ controls the proportion of positive and negative samples of each category. And we update those negative sample sets $\{\Omega_k\}_{k=1}^{k=K}$ per-epoch for efficiency.

Table 1: Summary of the dataset statistics

| Dataset | #Training | #Testing | #Classes | #Avg. Positive Classes |
|---------|-----------|----------|----------|------------------------|
| VOC     | 5,717     | 5,823    | 20       | 1.46                   |
| COCO    | 82,081    | 40,137   | 80       | 2.94                   |
| AWA     | 30,337    | 6,985    | 85       | 30.78                  |
| Ohsumed | 22,054    | 10,300   | 23       | 1.65                   |
| AAPD    | 53,840    | 1,000    | 54       | 2.41                   |

## 3.4 Model Training Summary

We describe the full training process of $\text{S}^2\text{ML}^2$-BBAM. To avoid inaccurate pseudo-labels in the early training stage, following [16], we warm up the classifier $f_{\mathbf{W}}(\cdot)$ with the BAM loss of Eq.6 over labeled samples $\mathcal{D}_l$ by $T_0$ epochs. Given the initialized $f_{\mathbf{W}}(\cdot)$, we continue to train it with the BBAM loss of Eq.8 over labeled samples $\mathcal{D}_l$ and unlabeled samples $\mathcal{D}_u$ by $T_t$ epochs. At each epoch, we update pseudo labels $\{\mathbf{y}_i^u\}_{i=1}^{i=N_u}$ by using Eq.5, label prototypes $\{\mathbf{c}_k\}_{k=1}^{k=K}$, $\{(\mu_k^{(p)}, (\sigma_k^2)^{(p)})\}_{k=1}^{k=K}$ and $\{(\mu_k^{(n)}, (\sigma_k^2)^{(n)})\}_{k=1}^{k=K}$ by using Eqs.9 and 10, and perform the negative sampling by using Eq.11. For clarity, the full training process is outlined in Appendix B.

# 4 Experiments

## 4.1 Experimental Settings

**Datasets.** We employ 5 widely used MLL datasets, including image datasets Pascal VOC-2012 (VOC) [27], MS-COCO2014 (COCO) [28] and Animals with Attributes2 (AWA) [29], text datasets Ohsumed [30] and AAPD [31]. For clarity, the detailed characteristics of these datasets are displayed in Table 1. Following [16], we transform these datasets into SSL versions. For each dataset, we randomly select $\pi$ training samples as labeled ones, and the remaining as unlabeled ones. We set $\pi \in \{5\%, 10\%, 15\%, 20\%\}$, to explore the performance of our method under different data proportions. The image size is resized to 224 for all datasets.

**Baselines.** We employ 5 baseline methods for comparisons, including SoftMatch [32], FlatMatch [33], MIME [34], DRML [15], and CAP [16]. DRML and CAP are SSMLL methods; SoftMatch and FlatMatch are SSL methods; MIME is a single-positive multi-label learning (SPMLL) method. For SSL and SPMLL methods, we follow CAP to apply them to SSMLL tasks.

**Evaluation metrics.** We employ 5 evaluation metrics, including Micro-F1, Macro-F1, mean average precision (mAP), Hamming Loss and One Loss [1], and compute them with the Scikit-Learn tool.[2]

**Implementation details.** We use the pre-trained ResNet-50 [35] as the backbone for image datasets and BERT-base-uncased model [36] for text datasets. We set the decay of EMA as 0.9997. The batch size is 32 for VOC, 128 for AWA and 64 for COCO, Ohsumed and AAPD. The warm-up epoch $T_0$ is 12. The $s$ and $m$ are 20 and 0.4 in VOC, 20 and 0.3 in COCO, 10 and 0.2 in AWA, Ohsumed and AAPD. The parameters for negative sampling $\eta$ are set to 5.

## 4.2 Results

The experimental results are presented in Table 2 and Table 3. Overall, our method achieves good performance on all metrics. Our model ranks *1st* on average on five datasets and has a significant advantage over baselines. The detailed analyses are presented as follows.

**Comparing with SSMLL methods:** We can observe that $\text{S}^2\text{ML}^2$-BBAM has advantages over recent SSMLL methods. Especially in the Micro-F1 and Macro-F1, our method has significant improvement. On both VOC and COCO, our F1 and mAP values increase by an average of 0.1 and 0.01. Furthermore, on Ohsumed and AAPD, we surprised to discover from the results that our method also has good results. In all data proportions, the average improvement on the mAP is 0.11, 0.14 on Macro-F1 and 0.19 on Micro-F1. This result is foreseeable because our method balanced angle variance using

---

[2]https://scikit-learn.org/stable/

Table 2: Experimental results on images datasets. The best results are highlighted in boldface.

| | VOC | | | | | | | | | | | | | | | | | | | |
|---|---|---|---|---|---|---|---|---|---|---|---|---|---|---|---|---|---|---|---|---|
| Method | Micro-F1↑ | | | | Macro-F1↑ | | | | mAP↑ | | | | Hamming Loss↓ | | | | One Loss↓ | | | |
| | $\pi=5\%$ | $\pi=10\%$ | $\pi=15\%$ | $\pi=20\%$ | $\pi=5\%$ | $\pi=10\%$ | $\pi=15\%$ | $\pi=20\%$ | $\pi=5\%$ | $\pi=10\%$ | $\pi=15\%$ | $\pi=20\%$ | $\pi=5\%$ | $\pi=10\%$ | $\pi=15\%$ | $\pi=20\%$ | $\pi=5\%$ | $\pi=10\%$ | $\pi=15\%$ | $\pi=20\%$ |
| SoftMatch | 0.6542 | 0.7187 | 0.7461 | 0.7484 | 0.5868 | 0.6630 | 0.6931 | 0.6876 | 0.6295 | 0.7235 | 0.7721 | 0.7867 | 0.0594 | 0.0368 | 0.0319 | 0.0294 | 0.4398 | 0.1655 | 0.1308 | 0.1148 |
| FlatMatch | 0.6493 | 0.7038 | 0.7420 | 0.7465 | 0.5344 | 0.6313 | 0.6666 | 0.6597 | 0.6468 | 0.7430 | 0.7923 | 0.8022 | 0.0386 | 0.0322 | 0.0313 | 0.0290 | 0.1983 | 0.1366 | 0.1238 | 0.1097 |
| MIME | 0.3650 | 0.6607 | 0.7013 | 0.7021 | 0.2439 | 0.5442 | 0.6425 | 0.5898 | 0.6653 | 0.7553 | 0.8090 | 0.8137 | 0.0546 | 0.0407 | 0.0336 | 0.0333 | 0.2099 | 0.1218 | 0.0835 | 0.0949 |
| DRML | 0.6450 | 0.6525 | 0.7274 | 0.7525 | 0.5660 | 0.5339 | 0.6864 | 0.7495 | 0.6058 | 0.6852 | 0.7131 | 0.7272 | 0.0564 | 0.0518 | 0.0377 | 0.0381 | 0.3542 | 0.2888 | 0.1720 | 0.1512 |
| CAP | 0.6162 | 0.6573 | 0.6798 | 0.7073 | 0.5822 | 0.6308 | 0.6536 | 0.6636 | 0.7616 | 0.8216 | 0.8348 | **0.8460** | 0.0801 | 0.0675 | 0.0622 | 0.0591 | 0.1303 | 0.0918 | 0.0827 | **0.0755** |
| S²ML²-BBAM | **0.7897** | **0.8401** | **0.8443** | **0.8458** | **0.7306** | **0.8015** | **0.8124** | **0.8141** | **0.7866** | **0.8345** | **0.8454** | 0.8458 | **0.0310** | **0.0259** | **0.0243** | **0.0233** | **0.1087** | **0.0867** | **0.0817** | 0.0795 |

| | COCO | | | | | | | | | | | | | | | | | | | |
|---|---|---|---|---|---|---|---|---|---|---|---|---|---|---|---|---|---|---|---|---|
| Method | Micro-F1↑ | | | | Macro-F1↑ | | | | mAP↑ | | | | Hamming Loss↓ | | | | One Loss↓ | | | |
| | $\pi=5\%$ | $\pi=10\%$ | $\pi=15\%$ | $\pi=20\%$ | $\pi=5\%$ | $\pi=10\%$ | $\pi=15\%$ | $\pi=20\%$ | $\pi=5\%$ | $\pi=10\%$ | $\pi=15\%$ | $\pi=20\%$ | $\pi=5\%$ | $\pi=10\%$ | $\pi=15\%$ | $\pi=20\%$ | $\pi=5\%$ | $\pi=10\%$ | $\pi=15\%$ | $\pi=20\%$ |
| SoftMatch | 0.5763 | 0.6273 | 0.6487 | 0.6676 | 0.4283 | 0.5265 | 0.5493 | 0.5830 | 0.5624 | 0.6194 | 0.6395 | 0.6622 | 0.0235 | 0.0218 | 0.0211 | 0.0205 | 0.1293 | 0.0948 | 0.0844 | 0.0879 |
| FlatMatch | 0.5960 | 0.6389 | 0.6590 | 0.6720 | 0.4794 | 0.5341 | 0.5710 | 0.5870 | 0.5827 | 0.6335 | 0.6542 | 0.6654 | **0.0227** | 0.0213 | 0.0208 | 0.0203 | 0.1215 | 0.1002 | 0.0933 | 0.0878 |
| MIME | 0.2982 | 0.4378 | 0.4906 | 0.5323 | 0.2557 | 0.3731 | 0.4096 | 0.4545 | 0.5372 | 0.5991 | 0.6379 | 0.6633 | 0.0302 | 0.0265 | 0.0250 | 0.0236 | 0.1495 | 0.1110 | 0.0883 | 0.0799 |
| DRML | 0.6071 | 0.6226 | 0.6492 | 0.6486 | 0.5345 | 0.5604 | 0.5779 | 0.5867 | 0.5118 | 0.5461 | 0.6026 | 0.6177 | 0.0242 | 0.0240 | 0.0230 | 0.0223 | 0.1438 | 0.1288 | 0.1243 | 0.1039 |
| CAP | 0.5629 | 0.5657 | 0.5724 | 0.5696 | 0.5230 | 0.5306 | 0.5402 | 0.5416 | 0.6243 | 0.6736 | **0.6911** | **0.7041** | 0.0523 | 0.0512 | 0.0499 | 0.0558 | 0.1004 | **0.0841** | **0.0788** | **0.0726** |
| S²ML²-BBAM | **0.6830** | **0.7074** | **0.7150** | **0.7246** | **0.6144** | **0.6480** | **0.6594** | **0.6726** | **0.6354** | **0.6741** | 0.6886 | 0.7023 | 0.0230 | **0.0212** | **0.0206** | **0.0201** | **0.1000** | 0.0878 | 0.0824 | 0.0799 |

| | AWA | | | | | | | | | | | | | | | | | | | |
|---|---|---|---|---|---|---|---|---|---|---|---|---|---|---|---|---|---|---|---|---|
| Method | Micro-F1↑ | | | | Macro-F1↑ | | | | mAP↑ | | | | Hamming Loss↓ | | | | One Loss↓ | | | |
| | $\pi=5\%$ | $\pi=10\%$ | $\pi=15\%$ | $\pi=20\%$ | $\pi=5\%$ | $\pi=10\%$ | $\pi=15\%$ | $\pi=20\%$ | $\pi=5\%$ | $\pi=10\%$ | $\pi=15\%$ | $\pi=20\%$ | $\pi=5\%$ | $\pi=10\%$ | $\pi=15\%$ | $\pi=20\%$ | $\pi=5\%$ | $\pi=10\%$ | $\pi=15\%$ | $\pi=20\%$ |
| SoftMatch | 0.6992 | 0.6973 | 0.7024 | 0.7024 | 0.5476 | 0.5284 | 0.5524 | 0.5457 | 0.6368 | 0.6524 | 0.6494 | 0.6518 | 0.2160 | 0.2155 | 0.2132 | 0.2126 | 0.1580 | 0.08876 | 0.1494 | 0.1549 |
| FlatMatch | 0.6918 | 0.6977 | 0.6989 | 0.7013 | 0.5221 | 0.5487 | 0.5507 | 0.5636 | 0.6393 | 0.6459 | 0.6565 | 0.6577 | 0.2190 | 0.2167 | 0.2165 | 0.2164 | 0.1029 | 0.0936 | 0.1116 | 0.1162 |
| MIME | 0.1470 | 0.3889 | 0.4893 | 0.4090 | 0.0705 | 0.1830 | 0.2659 | 0.2327 | 0.3992 | 0.3803 | 0.4762 | 0.5265 | 0.3570 | 0.3290 | 0.3064 | 0.3012 | 0.1850 | 0.2091 | 0.1664 | 0.2004 |
| DRML | 0.6827 | 0.6856 | 0.6942 | 0.6893 | 0.5399 | 0.5541 | 0.5727 | 0.5618 | 0.6160 | 0.6246 | 0.6377 | 0.6338 | 0.2285 | 0.2270 | 0.2226 | 0.2258 | 0.1360 | 0.1801 | 0.2609 | 0.1839 |
| CAP | 0.6868 | 0.7065 | 0.7091 | 0.7099 | 0.5742 | 0.5864 | 0.5905 | 0.5914 | 0.6390 | 0.6415 | 0.6440 | 0.6451 | 0.3120 | 0.2727 | 0.2589 | 0.2617 | 0.1146 | 0.0933 | 0.1045 | 0.1199 |
| S²ML²-BBAM | **0.7213** | **0.7255** | **0.7215** | **0.7279** | **0.5853** | **0.5914** | **0.5905** | **0.5944** | **0.6419** | 0.6463 | 0.6416 | 0.6476 | **0.2091** | **0.2060** | **0.2109** | **0.2042** | 0.1206 | 0.1103 | 0.1149 | 0.1188 |

Table 3: Experimental results on text datasets. The best results are highlighted in boldface.

| | Ohsumed | | | | | | | | | | | | | | | | | | | |
|---|---|---|---|---|---|---|---|---|---|---|---|---|---|---|---|---|---|---|---|---|
| Method | Micro-F1↑ | | | | Macro-F1↑ | | | | mAP↑ | | | | Hamming Loss↓ | | | | One Loss↓ | | | |
| | $\pi=5\%$ | $\pi=10\%$ | $\pi=15\%$ | $\pi=20\%$ | $\pi=5\%$ | $\pi=10\%$ | $\pi=15\%$ | $\pi=20\%$ | $\pi=5\%$ | $\pi=10\%$ | $\pi=15\%$ | $\pi=20\%$ | $\pi=5\%$ | $\pi=10\%$ | $\pi=15\%$ | $\pi=20\%$ | $\pi=5\%$ | $\pi=10\%$ | $\pi=15\%$ | $\pi=20\%$ |
| SoftMatch | 0.4769 | 0.4478 | 0.4462 | 0.4449 | 0.3056 | 0.2366 | 0.2348 | 0.2229 | 0.4664 | 0.5106 | 0.5218 | 0.5392 | 0.0756 | 0.0798 | 0.0801 | 0.0803 | 0.4213 | 0.5036 | 0.5274 | 0.5140 |
| FlatMatch | 0.5161 | 0.4836 | 0.4254 | 0.4472 | 0.3073 | 0.2262 | 0.1904 | 0.1775 | 0.4187 | 0.4751 | 0.4993 | 0.5139 | 0.0699 | 0.0747 | 0.0831 | 0.0799 | 0.3943 | 0.4416 | 0.5824 | 0.5008 |
| DRML | 0.3975 | 0.4015 | 0.4185 | 0.4055 | 0.1903 | 0.1972 | 0.1996 | 0.2070 | 0.1833 | 0.1931 | 0.2083 | 0.2140 | 0.0939 | 0.0868 | 0.0873 | 0.0851 | 0.6020 | 0.5677 | 0.5760 | 0.5496 |
| CAP | 0.5562 | 0.5776 | 0.5819 | 0.5455 | 0.4743 | 0.5144 | 0.5285 | 0.5214 | 0.4722 | 0.5370 | 0.5740 | 0.5995 | 0.0678 | 0.0840 | 0.0752 | 0.0967 | 0.3237 | 0.2746 | 0.2541 | 0.2493 |
| S²ML²-BBAM | **0.6671** | **0.7100** | **0.7196** | **0.7550** | **0.6058** | **0.6515** | **0.6719** | **0.7120** | **0.5537** | **0.6345** | **0.6604** | **0.6884** | **0.0467** | **0.0409** | **0.0243** | **0.0346** | **0.2417** | **0.2186** | **0.2068** | **0.1710** |

| | AAPD | | | | | | | | | | | | | | | | | | | |
|---|---|---|---|---|---|---|---|---|---|---|---|---|---|---|---|---|---|---|---|---|
| Method | Micro-F1↑ | | | | Macro-F1↑ | | | | mAP↑ | | | | Hamming Loss↓ | | | | One Loss↓ | | | |
| | $\pi=5\%$ | $\pi=10\%$ | $\pi=15\%$ | $\pi=20\%$ | $\pi=5\%$ | $\pi=10\%$ | $\pi=15\%$ | $\pi=20\%$ | $\pi=5\%$ | $\pi=10\%$ | $\pi=15\%$ | $\pi=20\%$ | $\pi=5\%$ | $\pi=10\%$ | $\pi=15\%$ | $\pi=20\%$ | $\pi=5\%$ | $\pi=10\%$ | $\pi=15\%$ | $\pi=20\%$ |
| SoftMatch | 0.3345 | 0.3325 | 0.3325 | 0.3279 | 0.0612 | 0.0514 | 0.0520 | 0.0481 | 0.3753 | 0.3949 | 0.4084 | 0.3990 | 0.0596 | 0.0598 | 0.0598 | 0.0602 | 0.6630 | 0.6630 | 0.6630 | 0.6627 |
| FlatMatch | 0.3221 | 0.3147 | 0.3155 | 0.3155 | 0.0519 | 0.0439 | 0.0437 | 0.0437 | 0.3571 | 0.3706 | 0.3570 | 0.3621 | 0.0607 | 0.0614 | 0.0613 | 0.0613 | 0.6629 | 0.6631 | 0.6635 | 0.6634 |
| DRML | 0.4160 | 0.4101 | 0.4027 | 0.4130 | 0.1024 | 0.1005 | 0.0998 | 0.1052 | 0.1465 | 0.1538 | 0.1579 | 0.1591 | 0.0545 | 0.0578 | 0.0521 | 0.0542 | 0.5450 | 0.5910 | 0.5280 | 0.5430 |
| CAP | 0.5722 | 0.5726 | 0.5504 | 0.5026 | 0.3917 | 0.4310 | 0.4257 | 0.4051 | 0.4095 | 0.4696 | 0.4899 | 0.4932 | 0.0432 | 0.0498 | 0.0571 | 0.0742 | 0.3010 | 0.2461 | 0.2523 | 0.2384 |
| S²ML²-BBAM | **0.7057** | **0.7279** | **0.7312** | **0.7316** | **0.5091** | **0.5825** | **0.5706** | **0.5823** | **0.5153** | **0.5903** | **0.5804** | **0.5930** | **0.0262** | **0.0241** | **0.0238** | **0.0238** | **0.1821** | **0.1500** | **0.1550** | **0.1590** |

Gaussian transformation, making the prediction of pseudo labels more accurate. Since Ohsumed and AAPD are text datasets, this result also demonstrates the good universality of our method.

**Comparing with SSL methods:** Our S²ML²-BBAM improves in both F1 and mAP metrics. For example, at $\pi = 5\%$, the mAP of S²ML²-BBAM is 0.07-0.16 higher than SoftMatch and 0.05-0.14 higher than FlatMatch across all datasets. We believe that this is because both methods are applied to multi classification tasks. So during the training process, it is more inclined to make single label classification decisions. Therefore, it doesn't perform as well as the SSMLL method. It can be inferred that it is important to set a dedicated method for SSMLL tasks.

**Comparing with SPMLL methods:** We observe that the performance of S²ML²-BBAM is better than MIME in all aspects. When $\pi = 5\%$, the average improvement on the mAP is 0.11, 0.42 on Macro-F1 and 0.41 on Micro-F1. We believe that this is because SPMLL is primarily designed to address the issue of incomplete labels. However, there is a large amount of unlabeled data in the setting of SSMLL tasks. This leads to the MIME method being unable to obtain single positive observation labels for these data, resulting in a significant loss of information. Therefore, the performance of MIME has declined.

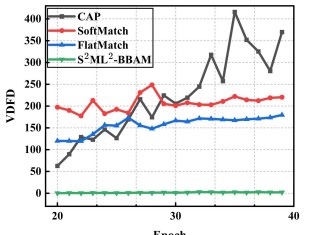 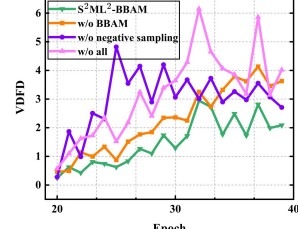

Figure 2: Comparison of VDFD on *VOC2012*.

Table 4: Results of the ablative study on *VOC2012* and *COCO*.

| | VOC | | | | | | | |
|---|---|---|---|---|---|---|---|---|
| Metric | $\pi = 5\%$ | | $\pi = 10\%$ | | $\pi = 15\%$ | | $\pi = 20\%$ | |
| | $S^2ML^2$-BBAM | w/o BBAM | $S^2ML^2$-BBAM | w/o BBAM | $S^2ML^2$-BBAM | w/o BBAM | $S^2ML^2$-BBAM | w/o BBAM |
| Micro-F1 | **0.7897** | 0.7845 | **0.8401** | 0.8206 | **0.8443** | 0.8301 | **0.8458** | 0.8318 |
| Macro-F1 | **0.7306** | 0.7247 | **0.8015** | 0.7789 | **0.8124** | 0.7988 | **0.8141** | 0.7967 |
| mAP | 0.7866 | **0.7881** | **0.8345** | 0.8204 | **0.8454** | 0.8274 | **0.8458** | 0.8282 |
| | COCO | | | | | | | |
| Metric | $\pi = 5\%$ | | $\pi = 10\%$ | | $\pi = 15\%$ | | $\pi = 20\%$ | |
| | $S^2ML^2$-BBAM | w/o BBAM | $S^2ML^2$-BBAM | w/o BBAM | $S^2ML^2$-BBAM | w/o BBAM | $S^2ML^2$-BBAM | w/o BBAM |
| Micro-F1 | **0.6830** | 0.6691 | **0.7074** | 0.6952 | **0.7150** | 0.7052 | **0.7246** | 0.7143 |
| Macro-F1 | **0.6144** | 0.5885 | **0.6480** | 0.6264 | **0.6594** | 0.6424 | **0.6726** | 0.6530 |
| mAP | **0.6354** | 0.5894 | **0.6741** | 0.6316 | **0.6886** | 0.6520 | **0.7023** | 0.6628 |

## 4.3 Ablation Study

To evaluate the effectiveness of the proposed BBAM loss, we perform several ablative studies by replacing it with the BAM loss (denoted by "w/o BBAM") on *VOC2012* and *COCO*. The results of the classification performance and VDFD are present in Table 4 and Fig.2, respectively. It clearly demonstrates that the proposed BBAM loss can significantly improve the classification performance and reduce variance differences between feature distributions. These results are expected because the BBAM loss can balance the variance bias between positive and negative samples from the perspective of the feature angle distribution for each label, leading to a fairer MLL classifier. Besides, we can observe that the VDFD of our $S^2ML^2$-BBAM is much lower than those SSMLL baselines during the training procedure, further proving the effectiveness of the BBAM loss in balancing the variance bias.

## 4.4 Parameter Evaluation

We conduct experiments on our method under different parameter settings. The experimental results are shown in Fig.3. We fix the $m$ value to 0.4 and set the $s$ values to $\{1, 10, 20, 30, 40, 50\}$ respectively. When $s$ is set between 1 and 10, the performance increases with the $s$. And when $s$ is set between 10 and 50, there is no significant change in the performance. One possible reason for this situation is that when the is small, the convergence speed of the model is too slow. So by the end of training, the model is not yet at its optimal state. We also explore the best accuracy by setting different cosine margins. We fix the value of $s$ to 20 and set the values of $m$ to $\{0.0, 0.2, 0.4, 0.6, 0.8, 1.0\}$ respectively. We find that the performance is at its optimum at $m = 0.4$.

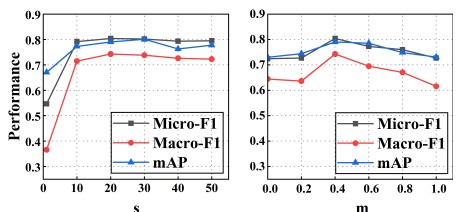

Figure 3: The sensitivity analysis of the rescaled norm and magnitude $\{s, m\}$ of cosine margin on VOC2012 with $\pi = 5\%$.

## 5 Related Works

**SSMLL methods.** Recently, SSMLL has received a lot of attention. In the latest methods, most focus on how to use the connections between labels to guild the training on unlabeled data. For instance, SMILE [37] calculates the relationships between labels by constructing adjacency graphs, while COIN [14] applies the well-known co-training method and learns under inductive settings. DRML [15] constructs the relationship network between labels by designing two classifiers and adopting

domain adaptation strategies. At the same time, how to effectively align pseudo labels with real labels is also an important issue. CAP [16] developed a class-distribution-aware thresholding strategy to control the assignment of positive and negative pseudo-labels. However, the current SSMLL methods have not paid attention to the **variance bias** problem, which affects the performance of the methods.

**MLL methods.** MLL has multiple research directions. Some methods focus on the model structure. For instance, [38] proposed a graph convolutional networks model to improve the performance of multi-label image recognition. [4] proposes a unified framework that combines CNNs and RNNs. Some others focus on exploiting label correlations to improve performance. LSF-CI[39] calculates instance correlation in the feature space and label correlation in the label space through a probabilistic neighborhood graph model and cosine similarity. Due to the complete label information of the training samples, the MLL method can theoretically achieve Bayesian optimal classifier boundaries. However, in semi supervised learning, incorrect pseudo labels may provide incorrect guidance for classification boundaries.

**SSL methods.** Pseudo Label [40] is one of the earliest semi-supervised learning methods for neural networks. It generates pseudo labels for unlabeled data and continuously improves the accuracy of pseudo labels as the model is optimized. As data augmentation technology has advanced, an increasing number of SSL methods are incorporating this technology [41, 42, 43, 44, 45, 46]. Further research has been conducted on the threshold issue of pseudo labels in [47, 48, 49]. By developing dynamic threshold strategies, they have been able to obtain more accurate pseudo labels, effectively enhancing the performance of the SSL methods. In order to utilize pseudo labels with low confidence but correct classification, [32] proposes an effective method that fits the confidence distribution of truncated Gaussian functions. Moreover, [33] discovered that the generalization ability of SSL models is impacted by disconnection between labeled data and unlabeled data, and proposed the FlatMatch method to address this issue. However, it's important to note that these SSL methods are designed to handle multi-class single-label tasks [50, 51] and cannot be directly applied to multi-label learning scenarios.

## 6  Conclusion

In this paper, we proposed a novel SSMLL method, namely $S^2ML^2$-BBAM. Our $S^2ML^2$-BBAM balances the variance bias between positive and negative samples from the perspective of the feature angle distribution for each label. To achieve this, we design a novel balanced binary angular margin loss by extending the traditional binary angular margin loss with feature angle distribution transformations under the Gaussian assumption, where the distributions are iteratively updated during classifier training. We also suggest an efficient prototype-based negative sampling method to maintain high-quality negative samples for each label. Empirical results demonstrate that our $S^2ML^2$-BBAM outperforms current SSMLL baseline methods.

## Limitations

From the empirical results, we found that $S^2ML^2$-BBAM suffers from slightly lower mAP scores on the benchmarks VOC and COOC when increasing the proportion of labeled training samples. This may restrict the range of applications and scenarios in which $S^2ML^2$-BBAM can be effectively used. And we will further exploit it in our future works.

## Broader Impacts

The paper focuses solely on the technical aspects of SSMLL algorithms. Therefore, this work can benefit a wide range of machine learning researchers. Also, we do not expect our efforts to have any negative consequences.

## Acknowledgements

We would like to acknowledge support for this project from the National Science and Technology Major Project of China (No.2021ZD0112500), the National Natural Science Foundation of China (No.62276113), and China Postdoctoral Science Foundation (No.2022M721321).

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

# A  Proof of Theoretical Results

*Proof.* Proof of Theorem 2.1. For any linear classifier $f(\mathbf{x}) = \text{sign}(\langle \mathbf{w}, \mathbf{x} \rangle + b)$, we first calculate its risk:

$\mathcal{R}_{ssl}(f)$

$= \mathbb{E}_{(\mathbf{x},y)\sim\mathfrak{D}}[\mathbb{1}(f(\mathbf{x}) \neq y)]$

$\propto \mathbb{E}_{(\mathbf{x},y)\sim\mathcal{D}^*}[\mathbb{1}(f(\mathbf{x}) \neq y)] + (1 - \epsilon_- - \epsilon_+)\mathbb{E}_{(\mathbf{x},y)\sim\mathcal{D}^*}[\mathbb{1}(f(\mathbf{x}) \neq y)] +$

$\quad \epsilon_-\mathbb{E}_{(\mathbf{x},-1)\sim\mathcal{D}^*}[\mathbb{1}(f(\mathbf{x}) \neq +1)] + \epsilon_+\mathbb{E}_{(\mathbf{x},+1)\sim\mathcal{D}^*}[\mathbb{1}(f(\mathbf{x}) \neq -1)]$

$= (2 - \epsilon_- - \epsilon_+)\mathbb{P}_{(\mathbf{x},y)\sim\mathcal{D}^*}[f(\mathbf{x}) \neq y] + \epsilon_-\mathbb{P}_{(\mathbf{x},-1)\sim\mathcal{D}^*}[f(\mathbf{x}) \neq +1] + \epsilon_+\mathbb{P}_{(\mathbf{x},+1)\sim\mathcal{D}^*}[f(\mathbf{x}) \neq -1]$

$= (2 - \epsilon_- - \epsilon_+) \cdot \big(\mathbb{P}[y = +1] \cdot \mathbb{P}[f(\mathbf{x}) = -1|y = +1] + \mathbb{P}[y = -1] \cdot \mathbb{P}[f(\mathbf{x}) = +1|y = -1]\big) +$

$\quad \epsilon_-\mathbb{P}[y = -1] \cdot \mathbb{P}[f(\mathbf{x}) = -1|y = -1] + \epsilon_+\mathbb{P}[y = +1] \cdot \mathbb{P}[f(\mathbf{x}) = +1|y = +1]$

$= (2 - \epsilon_- - \epsilon_+) \cdot \alpha \cdot \mathcal{R}(f, +1) + (2 - \epsilon_- - \epsilon_+) \cdot (1 - \alpha) \cdot \mathcal{R}(f, -1) +$

$\quad \epsilon_- \cdot (1 - \alpha) \cdot \mathbb{P}[f(\mathbf{x}) = -1|y = -1] + \epsilon_+ \cdot \alpha \cdot \mathbb{P}[f(\mathbf{x}) = +1|y = +1]$

where $\alpha = \mathbb{P}[y = +1]$, $\epsilon_+ = \mathbb{P}[\widehat{y} = -1|y = +1]$ and $\epsilon_- = \mathbb{P}[\widehat{y} = +1|y = -1]$.

Denote $\mathbf{x} = [x_1, \cdots, x_d]^\top$ and $\mathbf{w} = [w_1, \cdots, w_d]^\top$, we can explicitly calculate $\mathcal{R}(f, +1)$ and $\mathbb{P}[f(\mathbf{x}) = +1|y = +1]$ as:

$$\mathcal{R}(f, +1) = \mathbb{P}[f(\mathbf{x}) = -1|y = +1] = \mathbb{P}[\langle \mathbf{w}, \mathbf{x} \rangle + b < 0|y = +1] = \mathbb{P}[\sum_{i=1}^d w_i x_i + b < 0]$$

$$\mathbb{P}[f(\mathbf{x}) = +1|y = +1] = \mathbb{P}[\langle \mathbf{w}, \mathbf{x} \rangle + b > 0|y = +1] = \mathbb{P}[\sum_{i=1}^d w_i x_i + b > 0]$$

where $x_1, \cdots, x_d$ are *i.i.d.* drawn from Gaussian distributions $\{\mathcal{N}(\mu_i, (\sigma_+^i)^2)\}_{i=1}^d$ according to the definition of $\mathcal{P}^*$ in Eq.(2).

Similar to $\mathcal{R}(f, +1)$, we have

$$\mathcal{R}(f, -1) = \mathbb{P}[\sum_{i=1}^d w_i x_i + b > 0], \ \mathbb{P}[f(\mathbf{x}) = -1|y = -1] = \mathbb{P}[\sum_{i=1}^d w_i x_i + b < 0],$$

where $x_1, \cdots, x_d$ are *i.i.d.* drawn from Gaussian distributions $\{\mathcal{N}(-\mu_i, (\sigma_-^i)^2)\}_{i=1}^d$. Denote $f_{ssl}(\mathbf{x}) = \langle \mathbf{w}^*, \mathbf{x} \rangle + b^*$. According to the method of [52], we can prove $w_1^* = \cdots = w_d^* = 1$ by contradiction. Based on the properties of Gaussian distribution, $\mathcal{R}(f_{ssl}, +1)$, $\mathbb{P}[f_{ssl}(\mathbf{x}) = +1|y = +1]$, $\mathcal{R}(f_{ssl}, -1)$ and $\mathbb{P}[f_{ssl}(\mathbf{x}) = -1|y = -1]$ can be expressed as follows:

$$\mathcal{R}(f_{ssl}, +1) = \mathbb{P}[\sum_{i=1}^d x_i + b^* < 0] = \mathbb{P}\left[\frac{\sum_{i=1}^d (x_i - \mu_i)}{\sqrt{\sum_{i=1}^{i=d}(\sigma_+^{(i)})^2}} < \frac{-b^* - \sum_{i=1}^d \mu_i}{\sqrt{\sum_{i=1}^{i=d}(\sigma_+^{(i)})^2}}\right]$$

$$= \Phi\left(-\frac{b^* + \mu}{\Sigma}\right)$$

$$\mathbb{P}[f_{ssl}(\mathbf{x}) = +1|y = +1] = \mathbb{P}[\sum_{i=1}^d x_i + b^* > 0] = \mathbb{P}\left[\frac{\sum_{i=1}^d (x_i - \mu_i)}{\sqrt{\sum_{i=1}^{i=d}(\sigma_+^{(i)})^2}} > \frac{-b^* - \sum_{i=1}^d \mu_i}{\sqrt{\sum_{i=1}^{i=d}(\sigma_+^{(i)})^2}}\right]$$

$$= 1 - \Phi\left(-\frac{b^* + \mu}{\Sigma}\right)$$

$$\mathcal{R}(f_{ssl}, -1) = \mathbb{P}[\sum_{i=1}^d x_i + b^* > 0] = \mathbb{P}\left[\frac{\sum_{i=1}^d (x_i - (-\mu_i))}{\sqrt{\sum_{i=1}^{i=d}(\sigma_-^{(i)})^2}} > \frac{-b^* + \sum_{i=1}^d \mu_i}{\sqrt{\sum_{i=1}^{i=d}(\sigma_-^{(i)})^2}}\right]$$

$$= 1 - \Phi\left(\frac{-b^* + \mu}{M\Sigma}\right)$$

$$\mathbb{P}[f_{ssl}(\mathbf{x}) = -1|y=-1] = \mathbb{P}[\sum_{i=1}^{d} x_i + b^* < 0] = \mathbb{P}\left[\frac{\sum_{i=1}^{d}(x_i - (-\mu_i))}{\sqrt{\sum_{i=1}^{i=d}(\sigma_-^{(i)})^2}} < \frac{-b^* + \sum_{i=1}^{d}\mu_i}{\sqrt{\sum_{i=1}^{i=d}(\sigma_-^{(i)})^2}}\right]$$

$$= \Phi\left(\frac{-b^* + \mu}{M\Sigma}\right)$$

where $\Phi$ is c.d.f. of normal Gaussian distribution $\mathcal{N}(0,1)$. Then, we get

$$\mathcal{R}_{ssl}(f_{ssl}) = \alpha(2 - \epsilon_- - \epsilon_+)\Phi\left(-\frac{b^* + \mu}{\Sigma}\right) + (1-\alpha)(2 - \epsilon_- - \epsilon_+)\Phi\left(\frac{b^* - \mu}{M\Sigma}\right) +$$

$$(1-\alpha)\epsilon_-\Phi\left(\frac{-b^* + \mu}{M\Sigma}\right) + \alpha\epsilon_+\Phi\left(\frac{b^* + \mu}{\Sigma}\right)$$

We will find the optimal $b^*$ which minimizes the overall standard classification error $\mathcal{R}_{ssl}(f_{ssl})$ by taking $\frac{d\mathcal{R}_{ssl}(f_{ssl})}{db^*} = 0$. In detail, it is:

$$\frac{d\mathcal{R}_{ssl}(f_{ssl})}{db^*} = \alpha(2 - \epsilon_- - \epsilon_+)\frac{1}{\sqrt{2\pi}}\exp(-\frac{1}{2}(\frac{b^* + \mu}{\Sigma})^2)\frac{-1}{\Sigma} +$$

$$(1-\alpha)(2 - \epsilon_- - \epsilon_+)\frac{1}{\sqrt{2\pi}}\exp(-\frac{1}{2}(\frac{b^* - \mu}{M\Sigma})^2)\frac{1}{M\Sigma} +$$

$$(1-\alpha)\epsilon_-\frac{1}{\sqrt{2\pi}}\exp(-\frac{1}{2}(\frac{b^* - \mu}{M\Sigma})^2)\frac{-1}{M\Sigma} + \alpha\epsilon_+\frac{1}{\sqrt{2\pi}}\exp(-\frac{1}{2}(\frac{b^* + \mu}{\Sigma})^2)\frac{1}{\Sigma} = 0$$

which can be reformulated as:

$$(\frac{b^* + \mu}{\Sigma})^2 - (\frac{b^* - \mu}{M\Sigma})^2 = 2\log\left(\frac{M\alpha(2 - \epsilon_- - 2\epsilon_+)}{(1-\alpha)(2 - 2\epsilon_- - \epsilon_+)}\right)$$

Denote $B = \log\left(\frac{M\alpha(2-\epsilon_- -2\epsilon_+)}{(1-\alpha)(2-2\epsilon_- -\epsilon_+)}\right)$. Without loss of generality, we assume $B > 0$ and obtain:

$$(M^2 - 1)\Sigma^2(b^*)^2 + 2(M^2 + 1)\mu\Sigma^2 b^* + (M^2 - 1)\Sigma^2\mu^2 = 2BM^2\Sigma^4. \tag{12}$$

Consequently, $b^*$ can be given by selecting the smaller absolute value:

$$b^* = \begin{cases} \dfrac{-(M^2+1)\mu + 2M\mu\sqrt{1 + B\frac{(M^2-1)\Sigma^2}{2\mu^2}}}{M^2 - 1} & \text{if } M > 1, \\[4ex] \dfrac{-(M^2+1)\mu - 2M\mu\sqrt{1 + B\frac{(M^2-1)\Sigma^2}{2\mu^2}}}{M^2 - 1} & \text{if } M < 1, \end{cases}$$

Then when $M > 1$, the class-wise standard classification errors are:

$$\mathcal{R}(f_{ssl}, +1) = \Phi\left(A - M\sqrt{A^2 + q(M, \alpha, \epsilon_-, \epsilon_+)}\right),$$

$$\mathcal{R}(f_{ssl}, -1) = \Phi\left(-M \cdot A + \sqrt{A^2 + q(M, \alpha, \epsilon_-, \epsilon_+)}\right),$$

when $M < 1$, they are given by:

$$\mathcal{R}(f_{ssl}, +1) = \Phi\left(A + M\sqrt{A^2 + q(M, \alpha, \epsilon_-, \epsilon_+)}\right),$$

$$\mathcal{R}(f_{ssl}, -1) = \Phi\left(-M \cdot A - \sqrt{A^2 + q(M, \alpha, \epsilon_-, \epsilon_+)}\right),$$

where

$$A = \frac{2\mu}{(M^2 - 1)\Sigma}, \quad q(M, \alpha, \epsilon_-, \epsilon_+) = \frac{2\log\frac{M\alpha(2-\epsilon_- -2\epsilon_+)}{(1-\alpha)(2-2\epsilon_- -\epsilon_+)}}{M^2 - 1}.$$

When $\sum_{i=1}^{d}(\sigma_+^{(i)})^2 = \sum_{i=1}^{d}(\sigma_-^{(i)})^2 = \Sigma^2$, i.e. $M = 1$, Eq.(12) can be rewritten as:

$$4\mu b^* = 2\log\left(\frac{\alpha(2 - \epsilon_- - 2\epsilon_+)}{(1-\alpha)(2 - 2\epsilon_- - \epsilon_+)}\right)\Sigma^2.$$

In this case, $b^*$ can be expressed as follows:

$$b^* = \frac{\log\left(\frac{\alpha(2-\epsilon_- - 2\epsilon_+)}{(1-\alpha)(2-2\epsilon_- - \epsilon_+)}\right)\Sigma^2}{2\mu},$$

and corresponding class-wise standard classification errors are given by:

$$\mathcal{R}(f_{ssl}, +1) = \Phi\left(\frac{-2\mu^2 - \log\left(\frac{\alpha(2-\epsilon_- - 2\epsilon_+)}{(1-\alpha)(2-2\epsilon_- - \epsilon_+)}\right)\Sigma^2}{2\mu\Sigma}\right),$$

$$\mathcal{R}(f_{ssl}, -1) = \Phi\left(\frac{-2\mu^2 + \log\left(\frac{\alpha(2-\epsilon_- - 2\epsilon_+)}{(1-\alpha)(2-2\epsilon_- - \epsilon_+)}\right)\Sigma^2}{2\mu\Sigma}\right).$$

$\square$

*Proof.* Proof of Theorem 2.3. According to the results of Theorem 2.1, we can formulate the class-wise accuracy as:

$$p(+1) = 1 - \mathcal{R}(f_{ssl}, +1), \quad p(-1) = 1 - \mathcal{R}(f_{ssl}, -1).$$

Accordingly, the variance of class-wise accuracy can be expressed as:

$$\begin{aligned}
VCA(f_{ssl}) &= \mathrm{Var}(p(+1), p(-1)) = \mathrm{Var}(1 - \mathcal{R}(f_{ssl}, +1), 1 - \mathcal{R}(f_{ssl}, -1)) \\
&= \mathrm{Var}(\mathcal{R}(f_{ssl}, +1), \mathcal{R}(f_{ssl}, -1)) \\
&= \frac{(\mathcal{R}(f_{ssl}, +1) - \mathcal{R}(f_{ssl}, -1))^2}{2}.
\end{aligned}$$

For convenience, we assume $\log\left(\frac{\alpha(2-\epsilon_- - 2\epsilon_+)}{(1-\alpha)(2-2\epsilon_- - \epsilon_+)}\right) = 0$, and the conclusion will also hold when $M > \max\left(\frac{\alpha(2-\epsilon_- - 2\epsilon_+)}{(1-\alpha)(2-2\epsilon_- - \epsilon_+)}, 1\right)$ and $M < \min\left(\frac{\alpha(2-\epsilon_- - 2\epsilon_+)}{(1-\alpha)(2-2\epsilon_- - \epsilon_+)}, 1\right)$. When $M > 1$, it has $\mathcal{R}(f_{ssl}, -1) > \mathcal{R}(f_{ssl}, +1)$ because $q(M, \alpha, \epsilon_-, \epsilon_+) > 0$ and $A > 0$. Then according to Lagrange's Mean Value Theorem, there exists some $\xi$ such that

$$\begin{aligned}
&\mathcal{R}(f_{ssl}, -1) - \mathcal{R}(f_{ssl}, +1) \\
&= \Phi\left(-M \cdot A + \sqrt{A^2 + q(M, \alpha, \epsilon_-, \epsilon_+)}\right) - \Phi\left(A - M\sqrt{A^2 + q(M, \alpha, \epsilon_-, \epsilon_+)}\right) \\
&= \Phi'(\xi)\left(-M \cdot A + \sqrt{A^2 + q(M, \alpha, \epsilon_-, \epsilon_+)} - A + M\sqrt{A^2 + q(M, \alpha, \epsilon_-, \epsilon_+)}\right) \\
&= \frac{1}{\sqrt{2\pi}}\exp\left(-\frac{\xi^2}{2}\right)(M+1)\left(\sqrt{A^2 + q(M, \alpha, \epsilon_-, \epsilon_+)} - A\right).
\end{aligned}$$

By analyzing the variation of $q(M, \alpha, \epsilon_-, \epsilon_+)$, we can easily verify that $\mathcal{R}(f_{ssl}, -1) - \mathcal{R}(f_{ssl}, +1)$ is increasing when $M \to \infty$. Similarly, we can prove that $\mathcal{R}(f_{ssl}, +1) > \mathcal{R}(f_{ssl}, -1)$ when $M < 1$ and $\mathcal{R}(f_{ssl}, +1) - \mathcal{R}(f_{ssl}, -1)$ is increasing when $M \to 0$. $\square$

## B  The training procedure of the model

The *Algorithm* 1 provides a detailed description of the training process of the model.

---

**Algorithm 1** Training Procedure of $S^2 ML^2$-BBAM

---

**Input:**
 1: $\mathcal{D}_l$: the labeled training dataset
 2: $\mathcal{D}_u$: the unlabeled training dataset
 3: $T_0, T_t$: the number of warm-up epochs, the number of SSMLL training epochs
 4: $B_u$: the number of unlabeled batch size ;
**Output:** the classifier $f_{\mathbf{W}}(\cdot)$.
 5: **Initialize** the classifier parameter $\mathbf{W}$;
 6: Warm-up $f_{\mathbf{W}}(\cdot)$ on $\mathcal{D}_l$ with BAM loss Eq.(6) by $T_0$ epochs;
 7: **for** $t = 1$ **to** $T_t$ **do**
 8:     Calculate pseudo-labels $\{\mathbf{y}_i^u\}_{i=1}^{i=N_u}$ of $\mathcal{D}_u$ with Eq.(5);
 9:     Estimate $\{\mathbf{c}_k\}_{k=1}^{k=K}$, $\{(\mu_k^{(p)}, (\sigma_k^2)^{(p)})\}_{k=1}^{k=K}$ and $\{(\mu_k^{(n)}, (\sigma_k^2)^{(n)})\}_{k=1}^{k=K}$ with Eqs.(9) and (10);
10:     Construct $\{\Omega_k\}_{k=1}^{k=K}$ with Eq.(11);
11:     **for** $i = 1$ **to** $|\mathcal{D}_u| / B_u$ **do**
12:         Optimize $f_{\mathbf{W}}(\cdot)$ by minimizing the objective Eq.(4) with $\mathcal{D}_l, \mathcal{D}_u, \{\mathbf{y}_i^u\}_{i=1}^{i=N_u}$ and $\{\Omega_k\}_{k=1}^{k=K}$;
13:     **end for**
14: **end for**

---

## C  Time cost comparison

To examine the efficiency of $S^2 ML^2$-BBAM, we perform efficiency comparisons over our $S^2 ML^2$-BBAM, SSL baselines (SoftMatch and FlatMatch) and SSMLL baselines (DRML and CAP) on *VOC* and *COCO*. Table 5 shows the running time averaged 100 epochs. From Table 5, it can be seen that our method is competitive with the current SSMLL methods in the time efficiency and costs less time than the SSL baselines in practice.

Table 5: Time cost (second, **s**) of each training epoch on *VOC* and *COCO*.

| Method | VOC | COCO |
|---|---|---|
| SoftMatch | 79.3 | 726.2 |
| FlatMatch | 119.8 | 1658.1 |
| DRML | 4.9 | 30.4 |
| CAP | 28.4 | 312.5 |
| $S^2 ML^2$-**BBAM** | 33.1 | 276.3 |

