# OpenReview forum: "Semi-supervised Multi-label Learning with Balanced Binary Angular Margin Loss"
_NeurIPS.cc/2024/Conference — NeurIPS 2024 spotlight_

### Official Review · Reviewer_ZohP · 2024-07-07

**Soundness:** 3
**Presentation:** 3
**Contribution:** 3
**Rating:** 7
**Confidence:** 4

**Summary:**

In this paper, the authors investigate the topic of semi-supervised multi-label learning (SSMLL), and find an interesting problem in SSMLL, namely variance-bias issue, which means that the variance difference between positive and negative samples’ feature distributions for each label in SSMLL is much higher than ones in the supervised setting. And two novel theoretical analyses demonstrate that the variance bias will result in the unfairness of the induced multi-label classifier, leading to performance degradation. To address this issue, the authors propose a novel SSMLL method by balancing the variance bias from the feature angle distribution perspective with a new balanced binary angular margin (BBAM) loss. An efficient prototype-based negative sampling method is also suggested to maintain high-quality negative samples for each label. The authors conduct a number of experiments over both image and text benchmarks, and the results show the effectiveness of the proposed method.

**Strengths:**

- Interesting motivation: The paper is derived from a novel variance bias problem found by the authors. The authors investigate this problem and point out that it will result in an unfair classifier. A new and easy-to-implement BBAM loss is designed to balance the variance bias for each label and a novel SSMLL method is also proposed based on BBAM loss. The overall motivation is solid, and the proposed method is also reasonable.

- Well-written: The paper is well-written, with clear descriptions of the empirical phenomenon, theoretical analyses, method principles, and experimental design as well as results. The authors introduce the proposed BBAM loss and utilize techniques such as self-training and negative sampling to provide a viable solution, which is presented in a clear and concise manner.

- Experimental results over several image and text benchmarks demonstrate the effectiveness of the method: The paper provides experimental evidence of the proposed algorithm's effectiveness.

**Weaknesses:**

- Lack of quantitative analysis about the variance difference: The authors perform some extensive experiments on both image and text benchmarks and ablation study to show the effectiveness of the proposed BBAM loss and the corresponding method. It is also expected to perform experiments showing that the proposed BBAM loss can certainly eliminate the variance bias issue.
- Lack of experiments over benchmarks with more positive classes for each sample: The authors just perform experiments over some benchmarks, i.e., VOC, COCO, Ohsumed and AAPD, where each sample just consists of a few positive classes, and the average number of positive classes is 1~3. Some experiments are expected over benchmarks with more positive classes for each sample, such as AWA.
- Some minor typos: 1) “\alpha%” in line 212 should be “\alpha”. 2) The symbol of indicator function is inconsistent, such as Eq.(9) and Eq.(10). 3) “label angles and both positive …” in line 169 should be “label angles of both positive …”. 4) “Class-Aware Pesudo-” in line 144 should be “Class-Aware Pseudo-”.

**Questions:**

Please refer to the Weaknesses.

**Limitations:**

The authors have adequately addressed the limitations.

---

> ### Author Rebuttal · Authors · 2024-08-07
>
> First of all, we are very grateful for your time and effort in reviewing this submission. We are encouraged that you agree with the contributions of our paper. Below are the responses to your comments.
>
> **Q1: Lack of quantitative analysis about the variance difference.**
>
> **A1:** Yes, adding quantitative analysis about the variance difference will greatly strengthen the paper. Therefore, we compare the variance difference between feature distributions (VDFD) of positive and negative samples of our S$^2$ML$^2$-BBAM during the training procedure with ones of CAP, SoftMatch and FlatMatch, as well as the ablation variant of our S$^2$ML$^2$-BBAM by replacing BBAM with BAM (“w/o BBAM”) on label 6 of VOC2012. The experimental results are summarized as follows:
>
> | Epoch            | 22       | 24       | 26       | 28       | 30       | 32       | 34       | 36       | 38       | 40       |
> | ---------------- | -------- | -------- | -------- | -------- | -------- | -------- | -------- | -------- | -------- | -------- |
> | SoftMatch        | 177.43   | 182.69   | 184.28   | 248.53   | 200.90   | 203.44   | 210.61   | 214.32   | 218.90   | 220.28   |
> | FlatMatch        | 119.61   | 156.34   | 173.30   | 148.04   | 166.68   | 171.68   | 169.12   | 169.80   | 173.75   | 179.56   |
> | CAP              | 128.89   | 146.23   | 169.39   | 174.37   | 205.67   | 244.68   | 257.25   | 351.90   | 280.39   | 369.81   |
> | S$^2$ML$^2$-BBAM | **0.42** | **0.74** | **0.83** | **1.09** | **1.28** | **2.94** | **1.76** | **1.72** | **1.98** | **2.08** |
> | w/o BBAM         | 1.13     | 1.33     | 1.51     | 1.84     | 2.36     | 3.25     | 3.32     | 3.62     | 3.45     | 3.62     |
> |                  |          |          |          |          |          |          |          |          |          |          |
>
> From the results, the VDFD in the training procedure of our S$^2$ML$^2$-BBAM is much smaller than baselines, which demonstrates that S$^2$ML$^2$-BBAM can effectively balance the variance bias existing in current semi-supervised multi-label learning methods and obtain more fair performance. Otherwise, removing BBAM will increase the VDFD in the training procedure, and this shows the effectiveness of BBAM in balancing the variance bias.
>
> **Q2: Lack of experiments over benchmarks with more positive classes for each sample.**
>
> **A2:** We agree with you and add a new benchmark dataset, **Animals with Attributes2 (AWA)** [1], whose average number of positive classes is 30.78. Here are the experimental results on it:
>
> |                  | $\alpha$=5% |            |            |              |           | $\alpha$=10% |            |        |              |           |
> | ---------------- | ----------- | ---------- | ---------- | ------------ | --------- | ------------ | ---------- | ------ | ------------ | --------- |
> | Method           | Micro-F1    | Macro-F1   | mAP        | Hamming Loss | One Error | Micro-F1     | Macro-F1   | mAP    | Hamming Loss | One Error |
> | SoftMatch        | 0.6992      | 0.5476     | 0.6368     | 0.2160       | 0.1580    | 0.6973       | 0.5284     | 0.6524 | 0.2155       | 0.0887    |
> | FlatMatch        | 0.6918      | 0.5221     | 0.6393     | 0.2190       | 0.1029    | 0.6977       | 0.5487     | 0.6459 | 0.2167       | 0.0936    |
> | DRML             | 0.6827      | 0.5399     | 0.6160     | 0.2285       | 0.1360    | 0.6856       | 0.5541     | 0.6246 | 0.2270       | 0.1801    |
> | CAP              | 0.6868      | 0.5742     | 0.6390     | 0.3120       | 0.1146    | 0.7065       | 0.5864     | 0.6415 | 0.2727       | 0.0933    |
> | S$^2$ML$^2$-BBAM | **0.7213**  | **0.5853** | **0.6419** | **0.2091**   | 0.1206    | **0.7255**   | **0.5914** | 0.6463 | **0.2060**   | 0.1103    |
> |                  |             |            |            |              |           |              |            |        |              |           |
>
> |                  | $\alpha$=15% |            |        |              |           | $\alpha$=20% |            |        |              |           |
> | ---------------- | ------------ | ---------- | ------ | ------------ | --------- | ------------ | ---------- | ------ | ------------ | --------- |
> | Method           | Micro-F1     | Macro-F1   | mAP    | Hamming Loss | One Error | Micro-F1     | Macro-F1   | mAP    | Hamming Loss | One Error |
> | SoftMatch        | 0.7024       | 0.5524     | 0.6494 | 0.2132       | 0.1494    | 0.7024       | 0.5457     | 0.6518 | 0.2126       | 0.1549    |
> | FlatMatch        | 0.6989       | 0.5507     | 0.6565 | 0.2165       | 0.1116    | 0.7013       | 0.5636     | 0.6577 | 0.2164       | 0.1162    |
> | DRML             | 0.6942       | 0.5727     | 0.6377 | 0.2226       | 0.2609    | 0.6893       | 0.5618     | 0.6338 | 0.2258       | 0.1839    |
> | CAP              | 0.7091       | 0.5905     | 0.6440 | 0.2589       | 0.1045    | 0.7099       | 0.5914     | 0.6451 | 0.2617       | 0.1199    |
> | S$^2$ML$^2$-BBAM | **0.7215**   | **0.5905** | 0.6416 | **0.2109**   | 0.1149    | **0.7279**   | **0.5944** | 0.6476 | **0.2042**   | 0.1188    |
> |                  |              |            |        |              |           |              |            |        |              |           |
>
> In particular, our S$^2$ML$^2$-BBAM also performs better than baselines in most cases even when there are more positive classes for each sample.
>
> **Q3: Some minor typos.**
>
> **A3:** Thank you for your correction. We will update them and revise the manuscript.
>
>
>
> **Reference**
>
> [1] Lampert, C. H., H. Nickisch, S. Harmeling. Attribute-based classification for zero-shot visual object categorization. IEEE TPAMI, 36(3):453–465, 2013.

---

> > ### Comment · Reviewer_ZohP · 2024-08-14
> > **Official Comment by Reviewer ZohP**
> >
> > I have read the author's rebuttal and reviews from other reviewers. The authors addressed all my concerns, thus I decide to maintain my score.

---

> > > ### Author Response · Authors · 2024-08-14
> > >
> > > Thank you for your thoughtful suggestions and the positive feedback on our work.

---

### Official Review · Reviewer_MzwP · 2024-07-08

**Soundness:** 4
**Presentation:** 4
**Contribution:** 4
**Rating:** 7
**Confidence:** 5

**Summary:**

This paper focuses on the semi-supervised multi-label learning (SSMLL) task and propose a novel and interesting SSMLL method motivated by the variance bias problem, which implies that the variance difference of feature distributions of positive and negative samples for each label in SSMLL is much higher than one in the supervised learning setting. The authors present a theory showing that the variance bias will lead to an unfair classifier, and propose a new balanced binary angular margin (BBAM) loss to balance the variance bias issue from the perspective of feature angle distribution for each label. The performance of the proposed approach is validated on several image and text benchmark datasets, further confirming its superiority.

**Strengths:**

-	(Clarity) The paper is well organized and clearly written, the figures are also informative and well-designed.
-	(Novelty) The paper is thought-provoking! The variance bias issue within SSMLL and handling it by a new BBAM loss from from the perspective of feature angle distribution are highly innovative. Furthermore, the proposed techniques are also novel as far as I can tell.
-	(Quality) The paper is of good quality in my opinion. The algorithm is well designed for balancing the variance bias (via several label-specific linear Gaussian transformations) and efficient negative sampling (via prototype-based negative sample selection). The technical details are all correct as far as I can tell. The empirical evaluation is also comprehensive, covering 4 image and text datasets, and comparing against popular baselines.
-	(Significance) The proposed techniques are simple, easy to implement and experimentally highly effective, making the algorithm a strong, potentially impactful baseline for future researchers & practitioners to use and/or improve upon.

**Weaknesses:**

The linear Gaussian transformations and estimating label angle variances seem to be complex and may cost more time, which could pose challenges for practical applications. Thus, some efficiency experiments are expected to compare the real running time of the proposed method and other baselines, such as CAP. Finally, the reliance on the Gaussian assumption for feature distribution could limit the method's effectiveness in cases where this assumption does not hold.

**Questions:**

1. Is it feasible to apply the idea of balancing feature angle distributions in other fields, such as imbalanced or long-tailed setting?
2. Due to the prototype-based negative sampling method's computational demands, does it limit the scalability of the proposed method for other datasets including load of samples or categories?
3. The proposed BBAM loss relies on the Gaussian assumption for positive and negative feature distribution of each label. This may limit the proposed method’s effectiveness in cases where this assumption does not hold. Can the proposed method extend to other feature distribution cases? Or are there some more general feature distribution assumptions?

**Limitations:**

The authors have adequately addressed the limitations.

---

> ### Author Rebuttal · Authors · 2024-08-07
>
> Thank you for reviewing our paper. We are encouraged that you agree with the novelty and contributions of our paper. Below are the answers to your questions.
>
> **Q1: Some efficiency experiments are expected to compare the real running time of the proposed method and other baselines. Does it limit the scalability of the proposed method for other datasets including load of samples or categories?**
>
> **A1:** Thank you for your suggestion, we compare the real running time of our S$^2$ML$^2$-BBAM with the current semi-supervised multi-label learning method CAP and two semi-supervised learning methods SoftMatch and FlatMatch over VOC2012 and Animals with Attributes2 (AWA) [1], whose numbers of samples are 5717 and 30337, numbers of categories 20 and 85, average numbers of positive classes per instance 1.46 and 30.78. We report the average running time per epoch by running 100 epochs as follows:
>
> | Method           | Average running time per epoch (s) on VOC2012 | Average running time per epoch (s) on AWA |
> | ---------------- | --------------------------------------------- | ----------------------------------------- |
> | SoftMatch        | 79.3                                          | 268.9                                     |
> | FlatMatch        | 119.8                                         | 542.5                                     |
> | CAP              | 28.4                                          | 109.1                                     |
> | S$^2$ML$^2$-BBAM | 33.1                                          | 112.3                                     |
> |                  |                                               |                                           |
>
> As shown in the table, the running time of our S$^2$ML$^2$-BBAM is competitive with CAP, and much less than SoftMatch and FlatMatch. We kindly argue that the Gaussian linear transformation, estimating label angle variances and negaive sampling are simple, their additional computational cost is very few and they only perform one time per epoch. Overall, the scalability of our proposed S$^2$ML$^2$-BBAM is well and it can be adapted to other datasets including load of samples or categories. We will discuss the additional computational cost and show efficiency evaluations in the next version.
>
> **Q2: The reliance on the Gaussian assumption for feature distribution could limit the method's effectiveness in cases where this assumption does not hold.**
>
> **A2:** Thank you for your discussion. The Gaussian can be considered as a general tool for many cases. Besides, the linear transformation characteristics of Gaussian lead to simple calculations. We will investigate other distributions such as von Mise-Fisher distribution and GMM in our future works.
>
> **Q3: Is it feasible to apply the idea of balancing feature angle distributions in other fields, such as imbalanced or long-tailed settings?**
>
> **A3**: Thank you for your discussion. We kindly argue that the idea of balancing feature angle distributions can be applied to imbalanced or long-tailed settings, because balancing feature angle distributions can increase the diversity of minor classes and decrease the diversity of major classes to some extent, then address the imbalance issue. We will exploit it in our future works.
>
> **Reference**
>
> [1] Lampert, C. H., H. Nickisch, S. Harmeling. Attribute-based classification for zero-shot visual object categorization. IEEE TPAMI, 36(3):453–465, 2013.

---

> > ### Comment · Reviewer_MzwP · 2024-08-09
> >
> > Thanks for your responses. I have read them, and would like to keep my score.

---

> > > ### Author Response · Authors · 2024-08-12
> > >
> > > Thank you for your thoughtful suggestions and the positive feedback on our work.

---

### Official Review · Reviewer_zC26 · 2024-07-08

**Soundness:** 3
**Presentation:** 2
**Contribution:** 3
**Rating:** 7
**Confidence:** 4

**Summary:**

The paper proposed an interesting idea of using the balanced binary angular margin loss for semi-supervised multi-label learning learning (SSMLL). This is motivated by the empirical observation that the feature distributions of positive and negative samples for each label in SSMLL always suffer from the variance bias problem, and the theoretical results demonstrate that it potentially results in an unfair classifier for SSMLL. To address this issue, the authors propose to balance the variance bias between positive and negative samples, leading to a new and well-designed balanced binary angular margin loss. They also suggest a prototype-based negative sampling technique for efficient training. The idea of this paper is interesting and the proposed method is also well-motivated and well-supported with several experiments on both text and image benchmarks.

**Strengths:**

1. The idea is interesting, and the proposed technique is well motivated and clearly distinguished from prior works.
2. The variance bias problem is interesting for SSMLL, and the proposed theorems of the intra-class standard classification and the variance of class-wise accuracy can also indicate that the variance bias certainly results in an unfair classifier for SSMLL. This is very solid.
3. The proposed method is technically sound and it demonstrates a strong performance against current SSMLL methods, especially for Macro-F1 score. In addition, comprehensive experimental results on benchmark datasets clearly demonstrate the effectiveness of the proposed method.

**Weaknesses:**

1. How does the negative sampling affect the performance of the proposed method? Otherwise, the proportion of positive and negative samples of each category is set as 5. What happens if it is set to bigger or smaller values?
2. In the experiments, the ResNet-50 model was used as the backbone for image benchmarks. What is the backbone for text benchmarks?
3. Estimating label angle variances seem to complex and cost most time. Therefore, the efficiency experiments may be performed, such as comparing with the current CAP method.

**Questions:**

Please refer to the Weaknesses.

**Limitations:**

The authors have mostly addressed the limitations.

---

> ### Author Rebuttal · Authors · 2024-08-07
>
> First, we are very grateful for your time and effort in reviewing this paper. Below are the responses to your questions and comments.
>
> **Q1: How does the negative sampling affect the performance of the proposed method? What happens if the proportion of positive and negative samples of each category is set to bigger or smaller values?**
>
> **A1**: Thank you for your comment. We kindly argue that negative sampling is employed to select high-quality negative samples for each label as mentioned in the introduction and is a widely-used trick in multi-label learning [1,2,3,4]. Besides, we also perform a sensitivity analysis of the proportion ($\eta$) of positive and negative samples of each category over VOC2012 with $\alpha$=5% to show the effect of negative sampling on the classification performance.
>
> | $\eta$ | Micro-F1 | Macro-F1 | mAP    |
> | ------ | -------- | -------- | ------ |
> | 1      | 0.8007   | 0.7439   | 0.7980 |
> | 3      | 0.8007   | 0.7348   | 0.8030 |
> | 5      | 0.8055   | 0.7452   | 0.8004 |
> | 7      | 0.8037   | 0.7411   | 0.7958 |
> | 9      | 0.8024   | 0.7370   | 0.7876 |
> |        |          |          |        |
>
> As shown in the table, the bigger or smaller values of $\eta$ will result in a worse classification performance. One possible reason is that the bigger $\eta$ will introduce many false negative samples and the smaller one will cause the number of negative samples is not enough. We will present the corresponding results and ablation study of negative sampling in the next version, and exploit the selection of $\eta$ in our future work.
>
> **Q2: What is the backbone for text benchmarks?**
>
> **A2**: Thank you for your correction. We use the bert-base-uncased as the text backbone and will clarify it in the next version.
>
> **Q3: The efficiency experiments may be performed.**
>
> **A3**: Thank you for your suggestion, we compare the real running time of our S$^2$ML$^2$-BBAM with the current semi-supervised multi-label learning method CAP and two semi-supervised learning methods SoftMatch and FlatMatch over VOC2012 and Animals with Attributes2 (AWA) [5], whose numbers of samples are 5717 and 30337, numbers of categories 20 and 85, average numbers of positive classes per instance 1.46 and 30.78. We report the average running time per epoch by running 100 epochs as follows:
>
> | Method           | Average running time per epoch (s) on VOC | Average running time per epoch (s) on AWA |
> | ---------------- | ----------------------------------------- | ----------------------------------------- |
> | SoftMatch        | 79.3                                      | 268.9                                     |
> | FlatMatch        | 119.8                                     | 542.5                                     |
> | CAP              | 28.4                                      | 109.1                                     |
> | S$^2$ML$^2$-BBAM | 33.1                                      | 112.3                                     |
> |                  |                                           |                                           |
>
> As shown in the table, the running time of our S$^2$ML$^2$-BBAM is competitive with CAP, and much less than SoftMatch and FlatMatch. We kindly argue that the Gaussian linear transformation, estimating label angle variances and negaive sampling are simple, their additional computational cost is very few and they only perform one time per epoch. We will discuss the additional computational cost and show efficiency evaluations in the next version.
>
> **Reference**
>
> [1] Jiang, T., D. Wang, L. Sun, et al. Lightxml: Transformer with dynamic negative sampling for high-performance extreme multi-label text classification. In AAAI, pages 7987–7994. 2021.
>
> [2] Dahiya, K., D. Saini, A. Mittal, et al. Deepxml: A deep extreme multi-label learning framework applied to short text documents. In WSDM, pages 31–39. 2021.
>
> [3] Qaraei, M., R. Babbar. Meta-classifier free negative sampling for extreme multilabel classification. Machine Learning, pages 1–23, 2023.
>
> [4] Liu, Weiwei, et al. The emerging trends of multi-label learning. IEEE TPAMI, 44(11): 7955-7974,  2021.
>
> [5] Lampert, C. H., H. Nickisch, S. Harmeling. Attribute-based classification for zero-shot visual object categorization. IEEE TPAMI, 36(3):453–465, 2013.

---

> > ### Comment · Reviewer_zC26 · 2024-08-12
> >
> > I checked all the reviews and rebuttal. The authors do a good job to clarify my concerns. I would like to keep my scorings.

---

> > > ### Author Response · Authors · 2024-08-12
> > >
> > > Thank you for your thoughtful suggestions and the positive feedback on our work.

---

### Official Review · Reviewer_nsG4 · 2024-07-12

**Soundness:** 3
**Presentation:** 3
**Contribution:** 3
**Rating:** 6
**Confidence:** 3

**Summary:**

Based on the traditional binary loss function and negative sampling, when using labeled and pseudo-labeled samples for semi-supervised multi-label learning, there is an issue of variance bias between the feature distributions of positive and negative samples for each label. To solve this problem, authors balance the variance bias between positive and negative samples from the perspective of the feature angle distribution for each label. They also propose an efficient prototype-based negative sampling method to maintain high-quality negative samples for each label.

**Strengths:**

The definition of the problem scenario is accurate and clear. Their perspective on problem-solving is innovative. According to the experimental results, the proposed method is indeed effective in applications.

**Weaknesses:**

In terms of writing, the use of some mathematical symbols in the paper is not standardized. In terms of method analysis, the computation complexity of the proposed method is still unclear. In terms of experimentation, the evaluation metrics are not comprehensive enough.

**Questions:**

1. In experiments, authors describe the ratio of labeled samples in training as \alpha, and in Eq. (2), \alpha is also present, indicating probability. It is recommended that authors provide a more detailed explanation of mathematical symbols to avoid confusion.
2. In experiments, why were 6 algorithms compared in Table 1, while only 3 comparison algorithms were involved in Table 2?
3. Is evaluating the algorithm based on three metrics a bit limited in experiments?
4. In Parameter Evaluation Subsection, the description in 265-266 is confusing: "And when s is set between 10 and 50, there is no significant change in the performance. One possible reason for this situation is that when the is small, the convergence speed of the model is too slow." What does "the is small" mean? Is s between 1 and 10 considered smaller?
5. In terms of methodology, the proposed method trains a multi-label classifier based on binary loss on each label class. Does this overlook the important issue of label correlation in multi-label learning?

**Limitations:**

The authors mentioned that the limitation of the proposed method lies in the need to improve the experimental results in applications. I consider that whether or not the crucial issue of considering label correlation is effectively addressed is worth pondering.

---

> ### Author Rebuttal · Authors · 2024-08-07
>
> First, we would like to thank you for your time and effort in reviewing our submission. Next, we would like to respond to the main concerns raised in the comments.
>
> **Q1: About the some unstandardized mathematical symbols**
>
> **A1**: Thank you for your correction. We change the ratio of labeled samples in training to $\pi$ and will update it as well as introduce a detailed explanation of mathematical symbols as you suggested in the next version.
>
> **Q2: About the computation complexity**
>
> **A2**: Thank you for your comment, we compare the real running time of our S$^2$ML$^2$-BBAM with CAP and SoftMatch and FlatMatch over VOC2012 and Animals with Attributes2 (AWA) [1], whose numbers of samples are 5717 and 30337, numbers of categories 20 and 85, average numbers of positive classes per instance 1.46 and 30.78. We report the average running time per epoch by running 100 epochs as follows. As shown in the table, the running time of our S$^2$ML$^2$-BBAM is competitive with CAP, and much less than SoftMatch and FlatMatch. We kindly argue that the Gaussian linear transformation, estimating label angle variances and negative sampling are simple, their additional computational cost is very few and they only perform one time per epoch. The next version will discuss the additional computational cost and show efficiency evaluations.
>
> |Method|Average running time per epoch (s) on VOC|Average running time per epoch (s) on AWA|
> |---|---|---|
> |SoftMatch|79.3|268.9|
> |FlatMatch|119.8|542.5|
> |CAP|28.4|109.1|
> |S$^2$ML$^2$-BBAM|33.1|112.3|
>
> **Q3: About the evaluation metrics**
>
> **A3**: Thanks for your advice. We add two new metrics, including **Hamming Loss** and **One Error**. Then, the metrics used to evaluate the performance of multi-label classification cover *example-based* metrics (mAP, Hamming Loss, One Error) and *label-based* metrics (Micro-F1 and Macro-F1) [2]. The following two tables report the experimental results of Hamming Loss and One Error. As shown in two tables, our S$^2$ML$^2$-BBAM also performs better than baselines in most cases. The corresponding results will be updated in the next version.
>
> For Hamming Loss,
> ||VOC||||COCO||||
> |---|---|---|---|---|---|---|---|---|
> |Method|$\alpha$=5%|$\alpha$=10%|$\alpha$=15%|$\alpha$=20%|$\alpha$=5%|$\alpha$=10%|$\alpha$=15%|$\alpha$=20%|
> |SoftMatch|0.0594|0.0368 |0.0319|0.0294 |0.0235 |0.0218|0.0211|0.0205|
> |FlatMatch|0.0386 |0.0322|0.0313|0.0290|**0.0227** |0.0213|0.0208 |0.0203|
> |MIME|0.0546|0.0407|0.0336|0.0333|0.0302|0.0265 |0.0250|0.0236 |
> |DRML|0.0564 |0.0518 |0.0377|0.0381 |0.0242 |0.0240 |0.0230 |0.0223|
> |CAP|0.0801|0.0675|0.0622|0.0591|0.0523|0.0512 |0.0499 |0.0558|
> |S$^2$ML$^2$-BBAM|**0.0310**|**0.0259**|**0.0243**|**0.0233**|0.0230|**0.0212**|**0.0206**|**0.0201**|
>
> For One Error,
> ||VOC||||COCO||||
> |---|---|---|---|---|---|---|---|---|
> |Method|$\alpha$=5%|$\alpha$=10%|$\alpha$=15%|$\alpha$=20%|$\alpha$=5%|$\alpha$=10%|$\alpha$=15%|$\alpha$=20%|
> |SoftMatch|0.4398|0.1655|0.1308|0.1148|0.1293|0.0948|0.0844|0.0879|
> |FlatMatch|0.1983|0.1366|0.1238|0.1097|0.1215|0.1002|0.0933|0.0878|
> |MIME|0.2099|0.1218|0.0835|0.0949|0.1495|0.1110|0.0883|0.0799|
> |DRML|0.3542|0.2888|0.1720|0.1512|0.1438|0.1288|0.1243|0.1039|
> |CAP|0.1303|0.0918|0.0827|**0.0755**|0.1004|**0.0841**|**0.0788**|**0.0726**|
> |S$^2$ML$^2$-BBAM|**0.1087**|**0.0867**|**0.0817**|0.0795|**0.1000**|0.0878|0.0824|0.0799|
>
> **Q4: Only 3 comparison algorithms were involved in Table 2**
>
> **A4**: Thank you for your comment. We add SoftMatch and FlatMatch as baselines for text datasets, where Back-Translation (very time-consuming) is employed as the strong-augmentation method and none of the weak-augmentation methods is used following [3,4]. The corresponding results are reported in the following two tables. As shown in the two tables, our S$^2$ML$^2$-BBAM consistently outperforms two new baselines across all text benchmarks and metrics, demonstrating its effectiveness. We will update them in the next version.
>
> For Ohsumed,
> ||$\alpha$=5%|||||$\alpha$=10%|||||
> |---|---|---|---|---|---|---|---|---|---|---|
> |Method|Micro-F1|Macro-F1|mAP|HammingLoss|OneError |Micro-F1|Macro-F1|mAP|HammingLoss|OneError |
> |SoftMatch|0.4769|0.3056|0.4664|0.0756|0.4213|0.4478|0.2366|0.5106|0.0798|0.5036|
> |FlatMatch|0.5161|0.3073|0.4187|0.0699|0.3943|0.4836|0.2262|0.4751|0.0747|0.4416|
> |S$^2$ML$^2$-BBAM|**0.6671** |**0.6058**|**0.5537**|**0.0467**  |**0.2417**|**0.7100**  |**0.6515**|**0.6345**|**0.0409**  |**0.2186**|
>
> For AAPD,
> ||$\alpha$=5%|||||$\alpha$=10%|||||
> |---|---|---|---|---|---|---|---|---|---|---|
> |Method|Micro-F1|Macro-F1|mAP|HammingLoss|OneError|Micro-F1|Macro-F1|mAP|HammingLoss|OneError|
> |SoftMatch|0.3345|0.0612|0.3753|0.0596|0.6630|0.3325|0.0514|0.3949|0.0598|0.6630|
> |FlatMatch|0.3221|0.0519|0.3571|0.0607|0.6629|0.3147|0.0439|0.3706|0.0614|0.6631|
> |S$^2$ML$^2$-BBAM|**0.7057**|**0.5091**|**0.5153**|**0.0262**|**0.1821**|**0.7279**|**0.5825**|**0.5903**|**0.0241**|**0.1500**|
>
> **Q5: Confusing description in 265-266**
>
> **A5**: Thank you for your correction. We want to say that the performance is insensitive to the rescaled norm $s$ when $s\in [10, 50]$, and the best result is obtained when $s=20$. We will clarify it in the next version.
>
> **Q6: Overlook label correlation**
>
> **A6**: Thank you very much. We agree that label correlation is important to multi-label learning and we should present the overlook of label correlation in Limitations. The label correlation may be helpful to generate high-quality pseudo-labels for unlabeled instances, such as label propagation based on labeled instances and label corrections, as well as regularization term based on label correlation for the classifier weights. We will exploit them in our future work.
>
> **Reference**
>
> [1] Attribute-based classification for zero-shot visual object categorization. IEEE TPAMI, 2013.
>
> [2] A review on multi-label learning algorithms. IEEE TKDE, 2013.
>
> [4] SoftMatch: Addressing the Quantity-Quality Tradeoff in Semi-supervised Learning. ICLR, 2023.

---

> > ### Author Response · Authors · 2024-08-13
> >
> > Dear Reviewer nsG4,
> >
> > We kindly request your feedback or any additional questions, as our window for responding closes in 24 hours. We're more than happy to provide any clarification or further information you might need. Your input is valued, and we appreciate your time and consideration.

---

> > ### Comment · Reviewer_nsG4 · 2024-08-14
> >
> > Thanks to the hard work of the authors. My doubts and concerns have been clarified. I would like to upgrade my rating.

---

> > > ### Author Response · Authors · 2024-08-14
> > >
> > > Thank you for your thoughtful suggestions and the positive feedback on our work.

---

### Author Rebuttal · Authors · 2024-08-07

First of all, we sincerely thank all the reviewers for their great efforts in reviewing this submission and providing helpful and valuable comments. Since we cannot revise our paper during the rebuttal period, we plan to make the following revisions in our paper:

- According to most reviewers, we will revise the manuscript and present more experimental details as well as the comparison of the real running time.
- According to Reviewer nsG4, we will introduce new metrics (**Hamming Loss** and **One Error**) in Tables 1 and 2 and include the performance of SoftMatch and FlatMatch in Table 2.
- According to Reviewer zC26, we will include the ablation study of negative sampling and sensitivity analysis of the proportion ($\eta$) of positive and negative samples of each category.
- According to Reviewer ZoHp, we will add the quantitative analysis about the variance difference and a new benchmark dataset, **Animals with Attributes2 (AWA)** [1].

Besides, as suggested by Reviewer nsG4 and Reviewer Mzwp, we will consider introducing label correlation and other feature distribution assumptions and adapting balancing feature angle distributions in other fields as future work.

**Reference**

[1] Lampert, C. H., H. Nickisch, S. Harmeling. Attribute-based classification for zero-shot visual object categorization. IEEE TPAMI, 36(3):453–465, 2013.

---

### Public Comment · ~Changchun_Li1 · 2025-07-01
**Source code is released.**

Code for S$^2$ML$^2$-BBAM is available at https://github.com/changchunli/SSMLL-BBAM.

---

### Decision · Program_Chairs · 2024-09-25

**Decision:**

Accept (spotlight)

**Comment:**

All reviewers recommend the acceptance (3 accepts and 1 weak accept), therefore I also believe the paper must be accepted. The authors argue that learning with labeled and pseudo-labeled samples can result in the variance bias problem between the feature distributions of positive and negative samples for each label. To avoid this problem, the paper introduces as method to balance the variance bias between positive and negative samples from the perspective of the feature angle distribution for each label. The authors revised the paper based on the reviewer comments and the reviewers are satisfied with the responses.